# ON THE IMPACT OF ADVERSARIALLY ROBUST MODELS ON ALGORITHMIC RECOURSE

## ABSTRACT

The widespread deployment of machine learning models in various high-stakes settings has underscored the need for ensuring that individuals who are adversely impacted by model predictions are provided with a means for recourse. To this end, several algorithms have been proposed in recent literature to generate recourses. Recent research has also demonstrated that the recourses generated by these algorithms often correspond to adversarial examples. This key finding emphasizes the need for a deeper understanding of the impact of adversarially robust models (which are designed to guard against adversarial examples) on algorithmic recourse. In this work, we make one of the first attempts at studying the impact of adversarially robust models on algorithmic recourse. We theoretically and empirically analyze the cost (ease of implementation) and validity (probability of obtaining a positive model prediction) of the recourses output by state-of-the-art algorithms when the underlying models are adversarially robust. More specifically, we construct theoretical bounds on the differences between the cost and the validity of the recourses generated by various state-of-the-art algorithms when the underlying models are adversarially robust vs. non-robust. We also carry out extensive empirical analysis with multiple real-world datasets to not only validate our theoretical results, but also analyze the impact of varying degrees of model robustness on the cost and validity of the resulting recourses. Our theoretical and empirical analyses demonstrate that adversarially robust models significantly increase the cost and reduce the validity of the resulting recourses, thereby shedding light on the inherent trade-offs between achieving adversarial robustness in predictive models and providing easy-to-implement and reliable algorithmic recourse.

## 1 INTRODUCTION

As machine learning (ML) models are increasingly being deployed in high-stakes domains such as banking, healthcare, and criminal justice, it becomes critical to ensure that individuals who have been adversely impacted (e.g., loan denied) by the predictions of these models are provided with a means for recourse. To this end, several techniques have been proposed in recent literature to provide recourses to affected individuals by generating counterfactual explanations which highlight what features need to be changed and by how much in order to flip a model's prediction (Wachter et al., 2017; Dhurandhar et al., 2018; Ustun et al., 2019a; Pawelczyk et al., 2020a; Karimi et al., 2021b;a; Verma et al., 2020)[1]. For instance, Wachter et al. (2017) proposed a gradient-based approach which returns the nearest counterfactual resulting in the desired prediction. Ustun et al. (2019a) proposed an integer programming-based approach to obtain actionable recourses for linear classifiers. More recently, Karimi et al. (2021b; 2020c) leveraged the causal structure of the underlying data for generating recourses (Barocas et al., 2020; Mahajan et al., 2019; Pawelczyk et al., 2020a).

Prior research has also theoretically and empirically analyzed the properties of the recourses generated by state-of-the-art algorithms. For instance, several recent works (Rawal et al., 2021; Pawelczyk et al., 2020a; Dominguez-Olmedo et al., 2022; Upadhyay et al., 2021b) demonstrated that the recourses output by state-of-the-art algorithms are not robust to small perturbations to input instances, underlying model parameters, and to the recourses themselves. More recently, Pawelczyk et al.

---

[1]The terms counterfactual explanations (Wachter et al., 2017), contrastive explanations (Karimi et al., 2020b), and recourse (Ustun et al., 2019a) have often been used interchangeably in prior literature.

(2022a) demonstrated that the recourses output by state-of-the-art algorithms are very similar to adversarial examples. This finding is critical because there have been several efforts in the literature on adversarial ML (Huang et al., 2011; Kurakin et al., 2016; Biggio & Roli, 2018) to build adversarially robust models that are not susceptible to adversarial examples. However, the impact of such models on the quality and the correctness of the recourses output by state-of-the-art algorithms remains unexplored. The aforementioned connections between adversarial examples and recourses underscore the need for a deeper investigation of the impact of adversarially robust models (which are designed to guard against adversarial examples) on algorithmic recourse. Such an investigation becomes particularly critical as the need for adversarial robustness of predictive models as well as the ability to obtain easy-to-implement and reliable recourses have often been touted as the cornerstones of trustworthy and safe ML both by prior research as well as recent regulations Voigt & Von dem Bussche (2017); Hamon et al. (2020). However, there is no prior work that investigates the relationship and/or the trade-offs between these two critical pillars of trustworthy and safe ML.

In this work, we address the aforementioned gaps by making the first ever attempt at studying the impact of adversarially robust models on algorithmic recourse. We theoretically and empirically analyze the cost (ease of implementation) and validity (probability of obtaining a positive model prediction) of the recourses output by state-of-the-art algorithms when the underlying models are adversarially robust. More specifically, we construct theoretical bounds on the differences between the cost and the validity of the recourses generated by various state-of-the-art algorithms (e.g., gradient-based (Wachter et al., 2017; Laugel et al., 2017) and manifold-based (Pawelczyk et al., 2020b) methods) when the underlying models are adversarially robust vs. non-robust (See Section 4). To this end, we first derive theoretical bounds on the differences between the weights (parameters) of adversarially robust vs. non-robust models and then leverage these to bound the differences in the costs and validity of the recourses corresponding to these two sets of models.

We also carried out extensive empirical analysis with multiple real-world datasets from diverse domains. This analysis not only validated our theoretical bounds, but also unearthed several interesting insights pertaining to the relationship between adversarial robustness of predictive models and algorithmic recourse. More specifically, we found that the cost differences between the recourses corresponding to adversarially robust vs. non-robust models increase as the degree of robustness of the adversarially robust models increases. We also observed that the validity of recourses worsens as the degree of robustness of the underlying models increases. We further probed these insights by visualizing the resulting recourses in low dimensions using t-SNE plots, and found that the number of valid recourses around a given instance reduces as the degree of robustness of the underlying model increases.

## 2 RELATED WORK

**Algorithmic Recourse.** Several approaches have been proposed in recent literature to provide recourses to affected individuals (Dhurandhar et al., 2018; Wachter et al., 2017; Ustun et al., 2019a; Van Looveren & Klaise, 2019; Pawelczyk et al., 2020a; Mahajan et al., 2019; Karimi et al., 2020a;c; Dandl et al., 2020). These approaches can be broadly categorized along the following dimensions Verma et al. (2020): *type of the underlying predictive model* (e.g., tree based vs. differentiable classifier), *type of access* they require to the underlying predictive model (e.g., black box vs. gradient access), whether they encourage *sparsity* in counterfactuals (i.e., only a small number of features should be changed), whether counterfactuals should lie on the *data manifold*, whether the underlying *causal relationships* should be accounted for when generating counterfactuals, and whether the output produced by the method should be *multiple diverse counterfactuals* or a single counterfactual. In addition, Rawal & Lakkaraju (2020) also studied how to generate global, interpretable summaries of counterfactual explanations. Some recent works also demonstrated that the recourses output by state-of-the-art techniques might not be robust, i.e., small perturbations to the original instance (Dominguez-Olmedo et al., 2021; Slack et al., 2021), the underlying model (Upadhyay et al., 2021a; Rawal et al., 2021), or the recourse (Pawelczyk et al., 2022c) itself may render the previously prescribed recourse(s) invalid. These works also formulated and solved minimax optimization problems to find *robust* recourses to address the aforementioned challenges.

**Adversarial Examples and Robustness.** Prior works have shown that complex machine learning models, such as deep neural networks, are vulnerable to small changes in input (Szegedy et al., 2013). This behavior of predictive models allows for generating adversarial examples (AEs) by

adding infinitesimal changes to input targeted to achieve adversary-selected outcomes (Szegedy et al., 2013; Goodfellow et al., 2014). Prior works have proposed several techniques to generate AEs using varying degrees of access to the model, training data, and the training procedure (Chakraborty et al., 2018). While gradient-based methods (Goodfellow et al., 2014; Kurakin et al., 2016) return the smallest input perturbations which flip the label as adversarial examples, generative methods (Zhao et al., 2017) constrain the search for adversarial examples to the training data-manifold. Finally, some methods (Cisse et al., 2017) generate adversarial examples for non-differentiable and non-decomposable measures in complex domains such as speech recognition and image segmentation. Prior works have shown that Empirical Risk Minimization (ERM) does not yield models that are robust to adversarial examples (Goodfellow et al., 2014; Kurakin et al., 2016). Hence, to reliably train adversarially robust models, Madry et al. (2017) proposed the adversarial training objective which minimizes the worst-case loss within some $\epsilon$-ball perturbation region around the input instances.

**Intersections between Adversarial ML and Model Explanations.** There has been a growing interest in studying the intersection of adversarial ML and model explainability (Hamon et al., 2020). Among all the existing works focusing on this intersection, two explorations are relevant to our work (Shah et al., 2021; Pawelczyk et al., 2022b). Shah et al. (2021) studied the interplay between adversarial robustness and post hoc explanations (Shah et al., 2021) and demonstrated that gradient-based feature attribution methods (e.g., vanilla gradients, gradient times input, integrated gradients, smoothgrad) may severely violate the primary assumption of attribution – features with higher attribution are more important for model prediction – in case of non-robust models. However, their results also demonstrate that such a violation does not occur when the underlying models are robust to $\ell_2$ and $\ell_{\text{inf}}$ input perturbations. More recently, Pawelczyk et al. (2022a) demonstrated that recourses generated by certain state-of-the-art methods are very similar to adversarial examples, and also argued that the methods proposed to output recourses and adversarial examples are designed with similar goals of changing the input minimally in order to achieve the desired outcome. While the aforementioned works explored the connections between adversarial ML and model explanations, none of these works focus on analyzing the impact of adversarially robust models on the recourses output by state-of-the-art algorithms.

## 3 PRELIMINARIES

**Notation.** In this work, we denote a classifier $f : \mathcal{X} \to \mathcal{Y}$ mapping features $\mathbf{x} \in \mathcal{X}$ to labels $y \in \mathcal{Y}$, where $\mathbf{x}$ is a $d$-dimensional feature vector. We define a non-linear activation function $\phi(\cdot)$ such that $f(\mathbf{x}) = \phi(h(\mathbf{x}))$, where $h(\mathbf{x})$ is the logits of the linear model of the form $h(\mathbf{x}){=}\mathbf{w}^T\mathbf{x}$, for parameter weights $\mathbf{w}$. In addition, we represent the non-robust and adversarially robust models using $f_{\text{NR}}(\mathbf{x})$ and $f_{\text{R}}(\mathbf{x})$. Below we describe the methodological frameworks used for comparing recourses generated from non-robust and adversarially robust models.

**Adversarially Robust models.** Despite the superior performance of machine learning (ML) models, they are susceptible to adversarial examples (AEs), i.e., inputs generated by adding infinitesimal perturbations to the original samples targeted to change prediction label (Agarwal et al., 2019). One standard approach to ameliorate this problem is training a model using adversarial training which minimizes the worst-case loss within some perturbation region (the perturbation model) (Kolter & Madry). In particular, for a classifier, $f$ parameterized by weights $\theta$, loss function $\ell(\cdot)$, and training data $\{\mathbf{x}_i, y_i\}_{i=\{1,2,\ldots,n\}} \in \mathcal{D}_{\text{train}}$, the optimization problem of minimizing the worst-case loss within $\ell_p-$norm perturbation with radius $\epsilon$ is:

$$\min_{\theta} \frac{1}{|\mathcal{D}_{\text{train}}|} \sum_{(x,y)\in\mathcal{D}_{\text{train}}} \max_{\delta\in\Delta_{p,\epsilon}} \ell(f_{\theta}(\mathbf{x}+\delta)), y), \tag{1}$$

where $\mathcal{D}_{\text{train}}$ denotes the training dataset and $\Delta_{p,\epsilon} = \{\delta : \|\delta\|_p \leq \epsilon\}$ is the $\ell_p$ ball with radius $\epsilon$ centered around sample $\mathbf{x}$. We use $p = \infty$ for our theoretical analysis resulting in a closed-form solution of the model parameters $\theta$.

**Algorithmic Recourse.** One of the ways in which recourse can be realized is by explaining to affected individuals what features in their profile need to change and by how much in order to obtain a positive outcome. Counterfactual explanations which essentially capture the aforementioned information can therefore be used to provide recourse. The terms "counterfactual explanations" and "algorithmic recourse" have, in fact, become synonymous in recent literature (Karimi et al., 2020b; Ustun et al.,

2019b; Venkatasubramanian & Alfano, 2020). More specifically, algorithms that try to find algorithmic recourses do so by finding a counterfactual $\mathbf{x}' = \mathbf{x} + \zeta$ that is closest to the original instance $\mathbf{x}$ and change the model's prediction $f(\mathbf{x} + \zeta)$ to the target label. Next, we describe three methods we use to understand the implications of adversarially robust models on algorithmic recourses.

**Score CounterFactual Explanations (SCFE).** Given the classifier $f(\mathbf{x}) = \phi(h(\mathbf{x}))$ and a distance function $d : \mathbb{R}^d \times \mathbb{R}^d \to \mathbb{R}_+$, Wachter et al. (2017) define the problem of generating a recourse $\mathbf{x}' = \mathbf{x} + \zeta$ for sample $\mathbf{x}$ by minimizing the following objective:

$$\underset{\mathbf{x}'}{\arg\min}(h(\mathbf{x}') - s)^2 + \lambda d(\mathbf{x}', \mathbf{x}), \tag{2}$$

where $s$ is the target score for $\mathbf{x}'$, $\lambda$ is the regularization coefficient, and $d(\cdot)$ is the distance between sample $\mathbf{x}$ and its counterfactual counterpart $\mathbf{x}'$.

**C-CHVAE.** Given a Variational AutoEncoder (VAE) model with encoder $\mathcal{E}_\gamma$ and decoder $\mathcal{G}_\theta$ trained on the original data distribution $\mathcal{D}_{\text{train}}$, C-CHVAE (Pawelczyk et al., 2020b) aims to generate recourses in the latent space $\mathcal{Z}$, where $\mathcal{E}_\gamma : \mathcal{X} \to \mathcal{Z}$. The encoder $\mathcal{E}_\gamma$ transforms a given sample $\mathbf{x}$ into a latent representation $\mathbf{z} \in \mathcal{Z}$ and the decoder $\mathcal{G}_\theta$ takes $\mathbf{z}$ as input and produces $\hat{\mathbf{x}}$ as similar as possible to $\mathbf{x}$. To this end, given a sample $\mathbf{x}$, C-CHVAE generates the recourse $\zeta$ using the following objective function:

$$\zeta^* = \underset{\zeta \in \mathcal{Z}}{\arg\min}\|\zeta\| \quad \text{such that} \quad f(\mathcal{G}_\theta(\mathcal{E}_\gamma(\mathbf{x}) + \zeta)) \neq f(\mathbf{x}), \tag{3}$$

where $\mathcal{E}_\gamma$ allows to search for counterfactuals in the data manifold and $\mathcal{G}_\theta$ projects the latent counterfactuals to the feature space.

**Growing Spheres Method (GSM).** While the above techniques directly optimize specific objective functions for generating counterfactuals, GSM (Laugel et al., 2017) uses a search-based algorithm to generate recourses by randomly sampling points around the original instance $\mathbf{x}$ until a sample with the target label is found. In particular, GSM first draws an $\ell_2$-sphere around a given instance $\mathbf{x}$, randomly samples point within that sphere, and checks whether any sampled points result in target prediction. Finally, they contract or expand the sphere until a (sparse) counterfactual is found and finally returned. GSM defines a minimization problem using a function $c : \mathcal{X} \times \mathcal{X} \to \mathbb{R}_+$, where $c(\mathbf{x}, \mathbf{x}')$ is the cost of moving from instance $\mathbf{x}$ to counterfactual $\mathbf{x}'$.

$$\mathbf{x}'^* = \underset{\mathbf{x}' \in \mathcal{X}}{\arg\min}\{c(\mathbf{x}, \mathbf{x}') \mid f(\mathbf{x}') \neq f(\mathbf{x})\}, \tag{4}$$

where $\mathbf{x}'$ is sampled from the $\ell_2$-ball around $\mathbf{x}$ such that $f(\mathbf{x}') \neq f(\mathbf{x})$, $c(\mathbf{x}, \mathbf{x}') = \|\mathbf{x}' - \mathbf{x}\|_2 + \gamma\|\mathbf{x}' - \mathbf{x}\|_0$, and $\gamma \in \mathbb{R}_+$ is the weight associated to the sparsity objective.

## 4 THEORETICAL ANALYSIS

Next, we carry out a detailed theoretical analysis to bound the cost and validity differences of recourses generated by state-of-the-art recourse methods when the underlying models are adversarially robust vs. non-robust, for the case of linear classifiers. More specifically, we compare the cost differences w.r.t. the recourses obtained using 1) gradient-based methods such as SCFE (Wachter et al., 2017)(Sec. 4.1.1) and 2) manifold-based methods such as C-CHVAE (Pawelczyk et al., 2020b) (Sec. 4.1.2). Finally, we show that the validity of algorithmic recourse generated using existing methods for robust models is lower compared to that of non-robust models (Sec. 4.2).

### 4.1 COST ANALYSIS

The cost of a generated algorithmic recourse is defined as the distance (e.g., $\ell_1$ or $\ell_2$ distance) between the input instance $\mathbf{x}$ and the counterfactual $\mathbf{x}'$ obtained using a state-of-the-art recourse finding method (Verma et al., 2020). Algorithmic recourses with lower costs are considered better since they enable minimal changes to input to achieve the desired outcome. Here, we theoretically analyze the cost difference of generating recourses using algorithmic recourse methods when the underlying models are non-robust and adversarially robust.

#### 4.1.1 GRADIENT-BASED METHOD: SCFE

Next, we carry out a detailed theoretical analysis to bound the cost and validity difference of recourses generated by state-of-the-art recourse methods when the underlying models are adversarially robust

vs. non-robust. We derive the lower and upper bound for the cost difference of recourses generated by the SCFE (Wachter et al., 2017) method when the underlying models are adversarially robust vs. non-robust. Following previous works (Garreau & Luxburg, 2020; Hardt & Ma, 2017; Pawelczyk et al., 2022b; Rosenfeld et al., 2020; Ustun et al., 2019b), we focus on locally linear model approximations as this lays the foundation for understanding non-linear model behavior. For the cost difference, we first define the closed-form solution for the optimal cost $\zeta^*$ required to generate a recourse.

**Definition 1.** *(Optimal Cost from Pawelczyk et al. (2022b)) For a given scoring function $f$ with weights $\mathbf{w}$ the SCFE method generates a recourse $\mathbf{x}'$ for an input $\mathbf{x}$ using cost $\zeta$ such that:*

$$\zeta^* = m\frac{\lambda}{\lambda + \|\mathbf{w}\|_2^2} \cdot \mathbf{w}, \tag{5}$$

*where $m = s - h(\mathbf{x}')$ is the target residual, $s$ is the target score for $\mathbf{x}$, $h(x)$ is a local linear score approximation, and $\lambda$ is a given hyperparameter.*

**Theorem 1.** *(Cost difference for SCFE) For a given instance $\mathbf{x}$, let $\mathbf{x}'_{\mathrm{NR}}$ and $\mathbf{x}'_{\mathrm{R}}$ be the recourse generated using Wachter's algorithm for non-robust and adversarially robust models. Then, for a normalized Lipschitz activation function $\sigma(\cdot)$, the cost difference for the recourse generated for both models can be bounded as:*

$$\lambda|\frac{m_{\mathrm{NR}}}{\|\mathbf{w}_{\mathrm{NR}}\|_2} - \frac{m_{\mathrm{R}}}{\|\mathbf{w}_{\mathrm{R}}\|_2}| \le \|\zeta^*_{\mathrm{NR}} - \zeta^*_{\mathrm{R}}\|_2 \le \frac{\lambda}{\lambda + \|\mathbf{w}_{\mathrm{NR}}\|^2}\|\mathbf{w}_{\mathrm{NR}}\| + \frac{\lambda}{\lambda + \|\mathbf{w}_{\mathrm{R}}\|^2}\|\mathbf{w}_{\mathrm{R}}\|, \tag{6}$$

*where $\mathbf{w}_{\mathrm{NR}}$ and $\mathbf{w}_{\mathrm{R}}$ are the weights of the non-robust and adversarially robust models, $\lambda$ is the regularization coefficient in Wachter's algorithm, $m_{NR}=s-h_{NR}(\mathbf{x}'), m_R=s-h_R(\mathbf{x}')$ are the target residuals for robust ($f_{\mathrm{R}}(\mathbf{x}) = \phi(h_{\mathrm{R}}(\mathbf{x}))$) and non-robust models ($f_{\mathrm{NR}}(\mathbf{x})=\phi(h_{\mathrm{NR}}(\mathbf{x}))$), respectively.*

*Proof Sketch.* We derive the cost difference of recourses generated for non-robust and adversarially robust models by comparing their optimal solutions. Similar to Pawelczyk et al. (2022b), the upper bound results follow from Cauchy-Schwartz and triangle inequality. In addition, we also leverage reverse triangle inequality to derive a lower bound for the recourse difference. The equality of Equation 6 entails that the upper bound of the recourse difference will have a tighter bound if the $\ell_2$-norms of the weights $\mathbf{w}_{\mathrm{R}}$ and $\mathbf{w}_{\mathrm{B}}$ are bounded, and the lower bound of the recourse difference will be tighter if the output score of the non-robust and adversarially robust models is similar for the given sample $\mathbf{x}$. See Appendix A.1 for the complete proof. □

### 4.1.2 MANIFOLD-BASED METHOD: C-CHVAE

We extend our analysis of bounding the cost difference of generated recourses using manifold-based methods for non-robust and adversarially robust models. In particular, we leverage C-CHVAE (Pawelczyk et al., 2020b) that leverages variational autoencoders to generate counterfactuals. For a fair comparison, we assume that both models use the same encoder $\mathcal{I}_\gamma$ and decoder $\mathcal{G}_\theta$ networks for learning the latent space of the given input space $\mathcal{X}$.

**Definition 2.** *(Bora et al. (2017)) An encoder model $\mathcal{E}$ is L-Lipschitz if $\forall \mathbf{z}_1, \mathbf{z}_2 \in \mathcal{Z}$, we have:*

$$\|\mathcal{E}(\mathbf{z}_1) - \mathcal{E}(\mathbf{z}_2)\|_p \le L\|\mathbf{z}_1 - \mathbf{z}_2\|_p. \tag{7}$$

Using Definition 7, we now derive the lower and upper bounds of the cost difference of recourses generated for non-robust and adversarially robust models.

**Theorem 2.** *(Cost difference for C-CHVAE) Let $\mathbf{z}_{\mathrm{NR}}$ and $\mathbf{z}_{\mathrm{R}}$ be the generated recourse from C-CHVAE (Pawelczyk et al., 2020b) method in the latent space using an L-Lipschitz generative model $\mathcal{G}(\cdot)$ for a non-robust and adversarially robust model. Then, by definition of C-CHVAE, $\mathbf{x}_{\mathrm{NR}}=\mathcal{G}(\mathbf{z}_{\mathrm{NR}})=\mathbf{x} + \zeta_{\mathrm{NR}}$ and $\mathbf{x}_{\mathrm{R}}=\mathcal{G}(\mathbf{z}_{\mathrm{R}})=\mathbf{x} + \zeta_{\mathrm{R}}$ are the corresponding recourses in the input space. The cost difference between the recourses can then be bounded as:*

$$L(r^*_{\mathrm{R}} - r^*_{\mathrm{NR}}) \le \|\zeta_{\mathrm{R}} - \zeta_{\mathrm{NR}}\|_p \le L(r^*_{\mathrm{R}} + r^*_{\mathrm{NR}}), \tag{8}$$

*where $L$ is the Lipschitz constant of the generative model, and $r^*_{\mathrm{NR}}$ and $r^*_{\mathrm{R}}$ be the corresponding radii chosen by the algorithm such that they successfully return a recourse for the non-robust and adversarially robust model.*

*Proof Sketch.* The proof follows from Definition 7 and the triangle inequality. It shows that the cost difference for generating recourses using C-CHVAE is bounded by the product of the Lipschitz constant of the generative model and the radii chosen by the C-CHVAE to generate counterfactuals for the underlying non-robust and adversarially robust models. See Appendix A.2 for detailed proof. □

## 4.2 VALIDITY ANALYSIS

The validity of a given recourse $\mathbf{x}'$ is defined as the probability that it results in the desired outcome (Verma et al., 2020), denoted by $Pr(f(\mathbf{x}') = 1)$. Below, we analyze the validity of the recourses by first deriving the upper bound of the difference in non-robust and adversarially robust model weights, and then use this lemma to show that the validity of non-robust model is higher than for the adversarially robust model.

**Lemma 1.** *(Difference between non-robust and adversarially robust model weights) For a given instance $\mathbf{x}$, let $\mathbf{w}_{NR}$ and $\mathbf{w}_R$ be weights of the non-robust and adversarially robust model. Then, for a normalized Lipschitz activation function $\sigma(\cdot)$, the difference in the weights $\Delta_{\mathbf{w}}$ can be bounded as:*

$$\|\Delta_{\mathbf{w}}\|_2 \leq n\eta(y\|\mathbf{x}^T\|_2 + \epsilon\sqrt{d}) \tag{9}$$

*where $\eta$ is the learning rate, $\epsilon$ is the $\ell_2$-norm perturbation ball, $y$ is the label for $\mathbf{x}$, $n$ is the total number of training epochs, and $d$ is the dimension of the input features.*

*Proof Sketch.* We derive the upper bound of the difference in non-robust and adversarially robust model weights, denoted by $\Delta_{\mathbf{w}}$, and show that it is proportional to the dimension of the input features times the $\ell_2$ perturbation ball around the sample $\mathbf{x}$. See Appendix A.3 for the detailed proof. □

Next, we show that the probability of a recourse action resulting in the desired outcome is greater for a non-robust model compared to that of the adversarially robust model.

**Theorem 3.** *(Validity Comparison) For a given instance $\mathbf{x} \in \mathbb{R}^d$ and desired target label denoted by unity, let $\mathbf{x}_R$ and $\mathbf{x}_{NR}$ be the counterfactuals for adversarially robust $f_R(\mathbf{x})$ and non-robust $f_{NR}(\mathbf{x})$ models respectively. Then, $Pr(f_{NR}(\mathbf{x}_{NR}) = 1) \geq Pr(f_R(\mathbf{x}_R) = 1)$ if $|f_{NR}(\mathbf{x}_R) - f_{NR}(\mathbf{x}_{NR})| \leq n\eta(y\|\mathbf{x}^T\|_2 + \epsilon\sqrt{d})\|\mathbf{x}_R\|$, where $\eta$ is the learning rate, $\epsilon$ is the $\ell_2$-norm perturbation ball, $y$ is the label for $\mathbf{x}$, and $n$ is the total number of training epochs.*

*Proof Sketch.* We derive the difference between the probability that a valid recourse exists for a non-robust and adversarially robust model. Using data inequalities and Cauchy-Schwartz, we show that the condition for the validity is dependent on the weight difference of the models (Lemma 1). See Appendix A.4 for the detailed proof. □

## 5 EXPERIMENTAL EVALUATION

In this section, we empirically analyze the impact of adversarially robust models on the cost and validity of recourses. First, we empirically validate our theoretical bounds on differences between the cost and validity of recourses output by state-of-the-art recourse generation algorithms when the underlying models are adversarially robust vs. non-robust. Second, we carry out further empirical analysis to assess the differences in cost and validity of the resulting recourses as the degree of the adversarial robustness of the underlying model changes on three real-world datasets.

### 5.1 EXPERIMENTAL SETUP

Here, we describe the datasets used for our empirical analysis along with the predictive models, algorithmic recourse generation methods, and the evaluation metrics.

**Datasets.** We use three real-world datasets for our experiments: 1) The *German Credit* dataset [2] comprises demographic (age, gender), personal (marital status), and financial (income, credit duration) features from 1000 credit applicants, with each sample labeled as "good" or "bad" depending on their credit risk. The task is to successfully predict if a given individual is a "good" or "bad" customer

---
[2] https://archive.ics.uci.edu/ml/datasets/statlog+(german+credit+data)

in terms of associated credit risk. 2) The *Adult* dataset [3] contains demographic (e.g., age, race, and gender), education (degree), employment (occupation, hours-per week), personal (marital status, relationship), and financial (capital gain/loss) features for 48,842 individuals. The task is to predict if an individual's income exceeds $50K per year. 3) The *COMPAS* dataset[4] has criminal records and demographics features for 18,876 defendants who got released on bail at the U.S state courts during 1990-2009. The dataset is designed to train a binary classifier to classify defendants into bail (i.e., unlikely to commit a violent crime if released) vs. no bail (i.e., likely to commit a violent crime).

**Predictive models.** We generate recourses for the non-robust and adversarially robust version of Logistic Regression (linear) and Neural Networks (non-linear) models. We use two linear layers with sigmoid activation functions as our predictor and set the number of nodes in the intermediate layers to twice the number of nodes in the input layer, which is the size of the input dimension in each dataset.

**Algorithmic Recourse Methods.** We analyze the cost and validity for non-robust and adversarially robust models w.r.t. three popular classes of recourse generation methods, namely, gradient-based (SCFE), manifold-based (C-CHVAE), and random search-based (GSM) methods (described in Sec. 3).

**Evaluation metrics.** To concretely measure the impact of adversarial robustness on algorithmic recourse, we analyze the difference between cost and validity metrics for recourses generated using non-robust and adversarially robust model. To quantify the cost, we measure the average cost incurred to act upon the prescribed recourses across all test-set instances, i.e., $\text{Cost}(\mathbf{x}, \mathbf{x}') = \frac{1}{|\mathcal{D}_{\text{test}}|}\|\mathbf{x} - \mathbf{x}'\|_2$, where $\mathbf{x}$ is the input and $\mathbf{x}'$ is its corresponding recourse. To measure validity, we compute the probability of the generated recourse resulting in the desired outcome, i.e., $\text{Validity}(\mathbf{x}, \mathbf{x}') = \frac{|\{x':f(x')=1 \cap x'=g(x,f)\}|}{|\mathcal{D}_{\text{test}}|}$, where $g(x, f)$ returns recourses for input $\mathbf{x}$ and predictive model $f$.

**Implementation details.** We train non-robust and adversarially robust predictive models from two popular model classes (logistic regression and neural networks) for all three datasets. In the case of adversarially robust models, we adopt the commonly used min-max optimization objective for adversarial training using varying degree of robustness, i.e., $\epsilon \in \{0, 0.02, 0.05, 0.10, 0.15, 0.20, 0.25, 0.3\}$. Note that the model trained with $\epsilon{=}0$ is the non-robust model. Following Ballet et al. (2019) and Erdemir et al. (2021), we pre-processed the input data by removing categorical features for efficient training of our models. We follow Pawelczyk et al. (2022b) to set the hyperparameters for the algorithmic recourse methods.

## 5.2 EMPIRICAL ANALYSIS

Next, we describe the experiments that we carried out to understand the impact of adversarial robustness of predictive models on algorithmic recourse. More specifically, we will discuss (1) empirical verification of our theoretical bounds, (2) empirical analysis of the differences between the costs of recourses corresponding to non-robust vs. adversarially robust models, and (3) empirical analysis to compare the validity of the recourses corresponding to non-robust vs. adversarially robust models.

**Empirical Verification of Theoretical Bounds.** We empirically validate our theoretical findings from Section 4 on real-world datasets. In particular, we first estimate the empirical bounds (RHS of Theorems 1-2) for each instance in the test set by plugging the corresponding values of the parameters in the theorems and compare them with the empirical estimates of the cost differences between recourses generated using gradient- and manifold-based recourse methods (LHS of Theorems 1-2). Figure 9 show the results obtained from the aforementioned analysis of cost differences. We observe that our bounds are tight, and the empirical estimates fall well within our theoretical bounds. Similarly, we observe that the validity of the non-robust model, as denoted by $Pr(f_{\text{NR}}(x) = 1)$ in Theorem 3, was higher than the validity of the adversarially robust model for all the test samples in Adult, German Credit, COMPAS datasets, following the condition in Theorem 3 ( > 90% samples for the three datasets ) for a large number of training iterations used for training adversarially robust models with $\epsilon \in \{0, 0.02, 0.05, 0.1, 0.15, 0.2, 0.25, 0.3\}$.

**Cost Analysis.** To analyze the impact of adversarial robustness on the cost of recourses, we compute the difference between the cost for obtaining a recourse using non-robust and adversarially robust model and plotted this difference for varying degrees of robustness $\epsilon$. Results in Figure 2 show a

---

[3]`https://archive.ics.uci.edu/ml/datasets/Adult/`
[4]`https://github.com/propublica/compas-analysis`

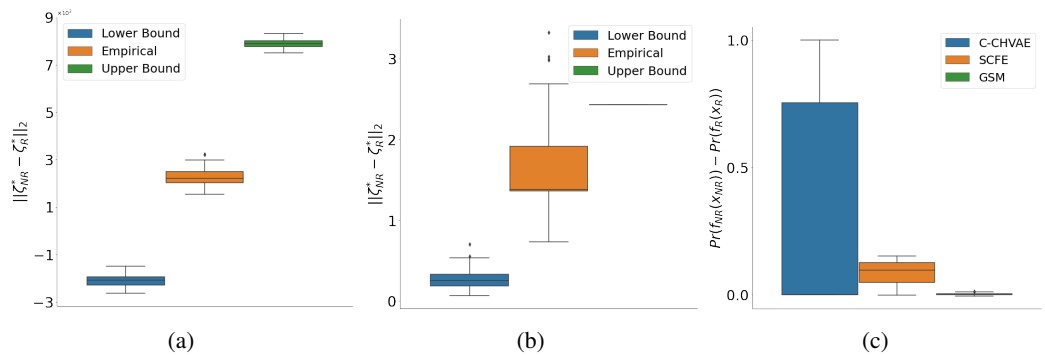

Figure 1: Empirically calculated cost differences (in orange) and our theoretical lower (in blue) and upper (in green) bounds for (a) C-CHVAE and (b) SCFE recourses corresponding to adversarially robust (trained using $\epsilon$=0.3) vs. non-robust models trained on the Adult dataset. Figure (c) is the empirical difference between the validity of recourses for non-robust and adversarially robust model. Results show no violations of our theoretical bounds. See Appendix B for results using different $\epsilon$ values.

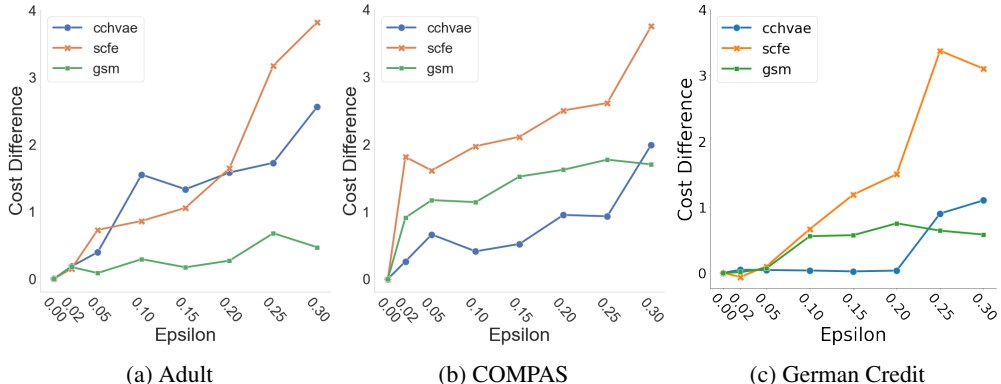

Figure 2: Analyzing cost differences between recourse generated using non-robust and adversarially robust neural networks for (a) Adult (b) COMPAS (c) German Credit datasets. We find that the cost difference (i.e., $\ell_2$−norm) between the recourses generated for non-robust and adversarially robust models increases for increasing values of $\epsilon$.

significant increase in incurred costs to find algorithmic recourse for adversarially robust models compared to the non-robust model for all the datasets with increasing degrees of robustness. We observe a similar trend for the case of logistic regression, as shown in Figure 6 in Appendix B. Further, we observe a relatively smoother increasing trend for cost differences in the case of SCFE compared to others, which can be attributed to the stochasticity present in C-CHVAE and GSM. We also observe a higher cost difference in SCFE for most datasets, which could result from the larger sample size used in C-CHVAE and GSM. We observe a similar trend in cost differences when the sample size per iteration is reduced, which also resulted in more iterations to find recourse.

**Validity Analysis.** To analyze the impact of adversarial robustness on the validity of recourses, we compute the fraction of recourses resulting in the desired outcome, generated using non-robust and adversarially robust model under resource constraints, and plot it against varying degrees of robustness $\epsilon$. Results in Figure 3 show that there is an even stronger impact of adversarial training on validity for neural networks trained on the three datasets. We observe a similar pattern for the case of the logistic regression model trained on the three datasets, shown in Appendix B. On average, we observe that the validity drops to zero for models adversarially trained with $\epsilon > 0.2$. To understand this further, we use t-SNE visualization (Van der Maaten & Hinton, 2008) – a non-linear dimensionality reduction technique – to map points in the dataset to two-dimensional space and demonstrate a gradual decline in valid recourses around a local neighborhood with increasing $\epsilon$ in Figure 4, where $x$ and $y$ be the names of reduced dimensions. This decline suggests that a large

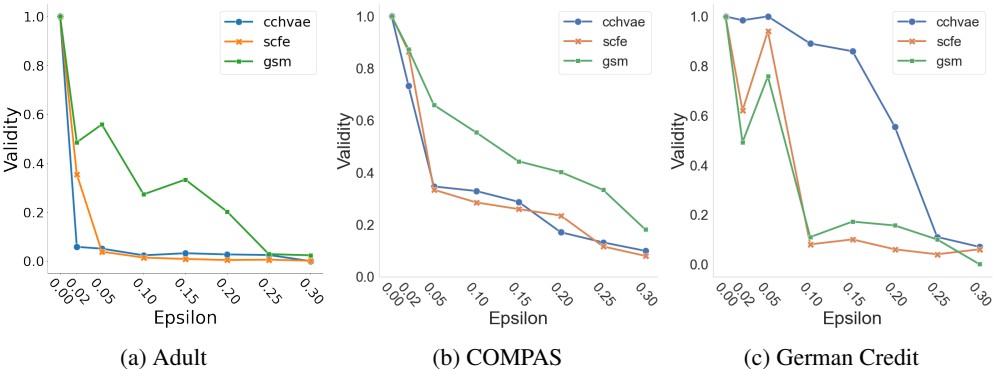

Figure 3: Analyzing validity of recourse generated using non-robust and adversarially robust neural networks for (a) Adult (b) COMPAS (c) German Credit datasets. We find that the validity decreases for increasing values of $\epsilon$.

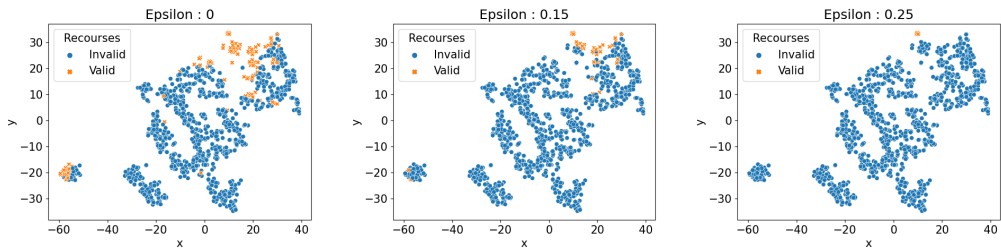

Figure 4: A t-SNE visualization of the change in availability of valid recourses (orange) for adversarially robust models trained using $\epsilon = [0, 0.15, 0.25]$, where a non-robust model is a model trained using $\epsilon = 0$. Results are shown for a neural network model trained on the Adult dataset. We observe fewer valid recourses for higher values of $\epsilon$ in this local neighborhood.

number of recourses in the neighborhood of the input sample are now being classified with the same class as the input. Hence, this supports our hypothesis that adversarially robust models severely impact the validity of recourses and make the recourse search computationally expensive.

## 6 CONCLUSION

In this work, we theoretically and empirically analyzed the impact of adversarially robust models on algorithmic recourse. We theoretically bounded the differences between the costs of the recourses output by two state-of-the-art counterfactual explanation methods (SCFE and C-CHVAE) when the underlying models are adversarially robust vs. non-robust. In addition, we also bounded the differences between the validity of the recourses corresponding to adversarially robust and non-robust models. We empirically validated our theoretical results using three real-world datasets (Adult, COMPAS, and German Credit) and two popular model classes (neural networks and logistic regression). Our theoretical and empirical analyses demonstrated that adversarially robust models significantly increase the cost and reduce the validity of the resulting recourses, thereby highlighting the inherent trade-offs between achieving adversarial robustness in predictive models and providing easy-to-implement and reliable algorithmic recourses. Our work also paves the way for several interesting future research directions at the intersection of algorithmic recourse and adversarial robustness in predictive models. For instance, given the aforementioned trade-offs, it would be interesting to develop novel techniques which enable end users to navigate these trade-offs based on their personal preferences – e.g., an end user may choose to sacrifice the adversarial robustness of the underlying model in order to secure lower cost recourses.

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

## A    PROOF FOR THEOREMS IN SECTION 4

Here, we provide detailed proofs of the Lemmas and Theorems defined in Section 4.

### A.1    PROOF FOR THEOREM 1

**Theorem 1.** *(Cost difference for SCFE) For a given instance* $\mathbf{x}$*, let* $\mathbf{x}'_{NR}$ *and* $\mathbf{x}'_R$ *be the recourse generated using Wachter's algorithm for the non-robust and adversarially robust models. Then, for a normalized Lipschitz activation function* $\sigma(\cdot)$*, the difference in the recourse for both models can be bounded as:*

$$\frac{\lambda}{\lambda+1}\varphi(s,\mathbf{x},\mathbf{w}_{NR},\mathbf{w}_R) \leq \|\zeta^*_{NR} - \zeta^*_R\|_2 \leq \left|\frac{\lambda}{\lambda+\|\mathbf{w}_{NR}\|^2}\right| \|\mathbf{w}_{NR}\| + \left|\frac{\lambda}{\lambda+\|\mathbf{w}_R\|^2}\right| \|\mathbf{w}_R\|, \quad (10)$$

*where* $\mathbf{w}_{NR}$ *and* $\mathbf{w}_R$ *are the weights of the non-robust and adversarially robust models,* $\lambda$ *is the regularization coefficient in Wachter's algorithm,* $\varphi(\cdot)$ *is a function that measures the shift in the model weights using the target and predicted scores.*

*Proof.* Following the definition of SCFE in Equation 2, we can find a counterfactual sample $\mathbf{x}'$ that is "closest" to the original instance $\mathbf{x}$ by minimizing the following objective:

$$\arg\min_{\mathbf{x}'}(h(\mathbf{x}') - y')^2 + \lambda d(\mathbf{x}', \mathbf{x}), \tag{11}$$

where $s$ is the target score, $\lambda$ is the regularization coefficient, and $d(\cdot)$ is the distance between the original and counterfactual sample $\mathbf{x}'$.

**Lower bound.** Using Lemma 1, the optimal cost for generating a valid recourse for a non-robust ($\zeta_{NR}^*$) and adversarially robust ($\zeta_R^*$) model can be written as:

$$\zeta_{\mathrm{NR}}^* = m_{\mathrm{NR}}\frac{\lambda}{\lambda + \|\mathbf{w}_{\mathrm{NR}}\|_2^2} \cdot \mathbf{w}_{\mathrm{NR}} \tag{12}$$

$$\zeta_{\mathrm{R}}^* = m_{\mathrm{R}}\frac{\lambda}{\lambda + \|\mathbf{w}_{\mathrm{R}}\|_2^2} \cdot \mathbf{w}_{\mathrm{R}}, \tag{13}$$

where $m_{\mathrm{NR}} = s - \mathbf{w}_{\mathrm{NR}}^T\mathbf{x}$ and $m_{\mathrm{R}} = s - \mathbf{w}_{\mathrm{R}}^T\mathbf{x}$. Subtracting and taking $\ell_2$-norm on both sides of Eqn. 12 and Eqn. 13, we get:

$$\|\zeta_{\mathrm{NR}}^* - \zeta_{\mathrm{R}}^*\|_2 = \left\|m_{\mathrm{NR}}\frac{\lambda}{\lambda + \|\mathbf{w}_{\mathrm{NR}}\|_2^2}\mathbf{w}_{\mathrm{NR}} - m_{\mathrm{R}}\frac{\lambda}{\lambda + \|\mathbf{w}_{\mathrm{R}}\|_2^2}\mathbf{w}_{\mathrm{R}}\right\|_2$$

$$\|\zeta_{\mathrm{NR}}^* - \zeta_{\mathrm{R}}^*\|_2 \geq \left|\left\|m_{\mathrm{NR}}\frac{\lambda}{\lambda + \|\mathbf{w}_{\mathrm{NR}}\|_2^2}\mathbf{w}_{\mathrm{NR}}\right\|_2 - \left\|m_{\mathrm{R}}\frac{\lambda}{\lambda + \|\mathbf{w}_{\mathrm{R}}\|_2^2}\mathbf{w}_{\mathrm{R}}\right\|_2\right|$$
$$\text{(Using reverse triangle inequality)}$$

$$\|\zeta_{\mathrm{NR}}^* - \zeta_{\mathrm{R}}^*\|_2 \geq \lambda\left|\left\|m_{\mathrm{NR}}\frac{1}{\lambda + \|\mathbf{w}_{\mathrm{NR}}\|_2^2}\mathbf{w}_{\mathrm{NR}}\right\|_2 - \left\|m_{\mathrm{R}}\frac{1}{\lambda + \|\mathbf{w}_{\mathrm{R}}\|_2^2}\mathbf{w}_{\mathrm{R}}\right\|_2\right|$$

$$\|\zeta_{\mathrm{NR}}^* - \zeta_{\mathrm{R}}^*\|_2 \geq \lambda\left|\left\|m_{\mathrm{NR}}\frac{1}{\|\mathbf{w}_{\mathrm{NR}}\|_2^2}\mathbf{w}_{\mathrm{NR}}\right\|_2 - \left\|m_{\mathrm{R}}\frac{1}{\|\mathbf{w}_{\mathrm{R}}\|_2^2}\mathbf{w}_{\mathrm{R}}\right\|_2\right| \qquad \text{(Since, } \lambda << \|\mathbf{w}\|_2)$$

$$\|\zeta_{\mathrm{NR}}^* - \zeta_{\mathrm{R}}^*\|_2 \geq \lambda\left|m_{\mathrm{NR}}\frac{1}{\|\mathbf{w}_{\mathrm{NR}}\|_2^2}\|\mathbf{w}_{\mathrm{NR}}\|_2 - m_{\mathrm{R}}\frac{1}{\|\mathbf{w}_{\mathrm{R}}\|_2^2}\|\mathbf{w}_{\mathrm{R}}\|_2\right|$$

$$\|\zeta_{\mathrm{NR}}^* - \zeta_{\mathrm{R}}^*\|_2 \geq \lambda\left|\frac{m_{\mathrm{NR}}}{\|\mathbf{w}_{\mathrm{NR}}\|_2} - \frac{m_{\mathrm{R}}}{\|\mathbf{w}_{\mathrm{R}}\|_2}\right|$$

**Upper bound.** Again, using the optimal recourse cost (Definition 1), we can derive the upper bound of the cost difference for generating recourses using non-robust and adversarially robust models:

$$\|\zeta_{\mathrm{NR}}^* - \zeta_{\mathrm{R}}^*\|_2 = \left\|\frac{(s - \mathbf{w}_{\mathrm{NR}}^T\mathbf{x})\lambda}{\lambda + \|\mathbf{w}_{\mathrm{NR}}\|^2} \cdot \mathbf{w}_{\mathrm{NR}} - \frac{(s - \mathbf{w}_{\mathrm{R}}^T\mathbf{x})\lambda}{\lambda + \|\mathbf{w}_{\mathrm{R}}\|^2} \cdot \mathbf{w}_{\mathrm{R}}\right\|$$

$$= \left\|\frac{(s - \mathbf{w}_{\mathrm{NR}}^T\mathbf{x})\lambda}{\lambda + \|\mathbf{w}_{\mathrm{NR}}\|^2} \cdot \mathbf{w}_{\mathrm{NR}} + \frac{(\mathbf{w}_{\mathrm{R}}^T\mathbf{x} - s)\lambda}{\lambda + \|\mathbf{w}_{\mathrm{R}}\|^2} \cdot \mathbf{w}_{\mathrm{R}}\right\|$$

$$\leq \left|\frac{(s - \mathbf{w}_{\mathrm{NR}}^T\mathbf{x})\lambda}{\lambda + \|\mathbf{w}_{\mathrm{NR}}\|^2}\right|\|\mathbf{w}_{\mathrm{NR}}\| + \left|\frac{(s - \mathbf{w}_{\mathrm{R}}^T\mathbf{x})\lambda}{\lambda + \|\mathbf{w}_{\mathrm{R}}\|^2}\right|\|\mathbf{w}_{\mathrm{R}}\| \qquad \text{(Using Triangle Inequality)}$$

Note that the difference between the target and the predicted score for both non-robust and adversarially robust models is upper bounded by a term that is always positive. Hence, we get:

$$\|\zeta_{\mathrm{NR}}^* - \zeta_{\mathrm{R}}^*\|_2 \leq \left|\frac{\lambda}{\lambda + \|\mathbf{w}_{\mathrm{NR}}\|^2}\right|\|\mathbf{w}_{\mathrm{NR}}\| + \left|\frac{\lambda}{\lambda + \|\mathbf{w}_{\mathrm{R}}\|^2}\right|\|\mathbf{w}_{\mathrm{R}}\|$$

$\square$

### A.2 PROOF FOR THEOREM 2

**Theorem 2.** *(Cost difference for C-CHVAE) Let $\mathbf{z}_{\mathrm{NR}}$ and $\mathbf{z}_{\mathrm{R}}$ be the solution returned by the C-CHVAE (Pawelczyk et al., 2020b) algorithmic recourse method by sampling from $\ell_p$-norm ball in*

*the latent space using an L-Lipschitz generative model $\mathcal{G}(\cdot)$ for a non-robust and adversarially robust model. By definition of the recourse method, let $\mathbf{x}_{\text{NR}} = \mathcal{G}(\mathbf{z}_{\text{NR}})$ and $\mathbf{x}_{\text{R}} = \mathcal{G}(\mathbf{z}_{\text{R}})$ be the corresponding recourses in the input space. The difference between them can then be bounded as:*

$$L(r_{\text{R}}^* - r_{\text{NR}}^*) \leq \|\mathbf{x}_{\text{R}} - \mathbf{x}_{\text{NR}}\|_p \leq L(r_{\text{R}}^* + r_{\text{NR}}^*), \tag{14}$$

*where $L$ is the Lipschitz constant of the generative model, and $r_{\text{NR}}^*$ and $r_{\text{R}}^*$ be the corresponding radii chosen by the algorithm such that they successfully return a recourse for the non-robust and adversarially robust model.*

*Proof.* From the formulation of the counterfactual algorithm, we can write the difference between $\mathbf{x}_{\text{R}}$ and $\mathbf{x}_{\text{NR}}$ as:

$$\|\mathbf{x}_{\text{R}} - \mathbf{x}_{\text{NR}}\|_p = \|\mathcal{G}_\theta(\mathbf{z}_{\text{R}}) - \mathcal{G}_\theta(\mathbf{z}_{\text{NR}})\|_p \tag{15}$$

**Lower bound.** Here, we present a lower bound on the $\ell_p$ norm of the cost difference between a baseline and robust model. Using Equation 15, we get:

$$
\begin{aligned}
\|\mathbf{x}_{\text{R}} - \mathbf{x}_{\text{NR}}\|_p &= \|\mathcal{G}_\theta(\mathbf{z}_{\text{R}}) - \mathcal{G}_\theta(\mathbf{z}) - \mathcal{G}_\theta(\mathbf{z}_{\text{NR}}) + \mathcal{G}_\theta(\mathbf{z})\|_p && \text{(16)} \\
&\geq \|\mathcal{G}_\theta(\mathbf{z}_{\text{R}}) - \mathcal{G}_\theta(\mathbf{z})\|_p - \|\mathcal{G}_\theta(\mathbf{z}_{\text{NR}}) - \mathcal{G}_\theta(\mathbf{z})\|_p && \text{(since } \|a-b\|_p \geq \|a\|_p - \|b\|_p\text{)} \\
&\geq L\|\mathbf{z}_{\text{R}} - \mathbf{z}\|_p - L\|\mathbf{z}_{\text{NR}} - \mathbf{z}\|_p && \text{(17)} \\
\|\mathbf{x}_{\text{R}} - \mathbf{x}_{\text{NR}}\|_p &\geq L(r_{\text{R}}^* - r_{\text{NR}}^*), && \text{(Using (Sidford))}
\end{aligned}
$$

where $\mathbf{z}$ is the latent space representation for the original point $\mathbf{x}$, $r_{\text{R}}^*$ and $r_{\text{NR}}^*$ are the radius of the $\ell_p$-norm for generating samples from the robust and baseline model. Note that using the radius of the $\ell_p$ norm in the above equation provides a tighter lower bound.

**Upper bound.** Using Equation 15, we can derive the upper bound using Lemma 1 and the triangle inequality.

$$
\begin{aligned}
\|\mathbf{x}_{\text{R}} - \mathbf{x}_{\text{NR}}\|_p &\leq \|\mathcal{G}_\theta(\mathbf{z}_{\text{R}}) - \mathbf{x}\|_p + \|\mathbf{x} - \mathcal{G}_\theta(\mathbf{z}_{\text{NR}})\|_p && \text{(Using triangle inequality)} \\
&= \|\mathcal{G}_\theta(\mathbf{z}_{\text{R}}) - \mathcal{G}_\theta(\mathbf{z})\|_p + \|\mathcal{G}_\theta(\mathbf{z}) - \mathcal{G}_\theta(\mathbf{z}_{\text{NR}})\|_p && \text{(18)} \\
&\leq L\|\mathbf{z}_{\text{R}} - \mathbf{z}\|_p + L\|\mathbf{z} - \mathbf{z}_{\text{NR}}\|_p && \text{(Using Lemma 1)} \\
\|\mathbf{x}_{\text{R}} - \mathbf{x}_{\text{NR}}\|_p &\leq L(r_{\text{R}}^* + r_{\text{NR}}^*), && \text{(19)}
\end{aligned}
$$

where $r_{\text{R}}^*$ and $r_{\text{NR}}^*$ is the radius of the $\ell_p$-norm for generating samples from the robust and baseline model, respectively. $\qquad\square$

### A.3 Proof for Lemma 1

**Lemma 1.** *(Difference between non-robust and adversarially robust model weights) For a given instance $\mathbf{x}$, let $\mathbf{w}_{\text{NR}}$ and $\mathbf{w}_{\text{R}}$ be weights of the non-robust and adversarially robust model. Then, for a normalized Lipschitz activation function $\sigma(\cdot)$, the difference in the weights $\Delta_{\mathbf{w}}$ can be bounded as:*

$$\|\Delta_{\mathbf{w}}\|_2 \leq n\eta(y\|\mathbf{x}^T\|_2 + \epsilon\sqrt{d}) \tag{20}$$

*where $\eta$ is the learning rate, $\epsilon$ is the $\ell_2$-norm perturbation ball, $y$ is the label for $\mathbf{x}$, $n$ is the total number of training epochs, and $d$ is the dimension of the input features.*

*Proof.* Without loss of generality, we consider the case of binary classification which uses the binary cross entropy or logistic loss. Let us denote the baseline and robust models as $f_{\text{NR}}(\mathbf{x}) = \mathbf{w}_{\text{NR}}^T\mathbf{x}$ and $f_{\text{R}}(\mathbf{x}) = \mathbf{w}_{\text{R}}^T\mathbf{x}$, where we have removed the bias term for simplicity. We consider the class label as $y \in \{+1, -1\}$, and loss function $\mathcal{L}(f(\mathbf{x})) = \log(1 + \exp(-y.f(\mathbf{x})))$. Note that an adversarially robust model $f_{\text{R}}(\mathbf{x})$ is commonly trained using a min-max objective, where the inner maximization problem is given by:

$$\max_{\|\delta\| \leq \epsilon} \mathcal{L}(\mathbf{w}_{\text{R}}^T(\mathbf{x} + \delta), y), \tag{21}$$

where $\delta$ is the adversarial perturbation added to a given sample $\mathbf{x}$ and $\epsilon$ denotes the the perturbation norm ball around $\mathbf{x}$. Since our loss function is monotonic decreasing, the maximization of the loss function applied to a scalar is equivalent to just minimizing the scalar quantity itself, i.e.,

$$\max_{\|\delta\| \leq \epsilon} \mathcal{L}\left(y \cdot (\mathbf{w}_{\text{R}}^T(\mathbf{x} + \delta))\right) = \mathcal{L}\left(\min_{\|\delta\| \leq \epsilon} y \cdot (\mathbf{w}_{\text{R}}^T(\mathbf{x} + \delta))\right) = \mathcal{L}\left(y \cdot (\mathbf{w}_{\text{R}}^T\mathbf{x}) + \min_{\|\delta\| \leq \epsilon} y \cdot \mathbf{w}_{\text{R}}^T\delta\right) \tag{22}$$

The optimal solution to $\min_{\|\delta\| \le \epsilon} y \cdot \mathbf{w}_{\mathrm{R}}^{\mathrm{T}} \delta$ is given by $-\epsilon \|\mathbf{w}_{\mathrm{R}}^{\mathrm{T}}\|_1$ (Kolter & Madry). Therefore, instead of solving the min-max problem for an adversarially robust model, we can convert it to a pure minimization problem, i.e.,

$$\min_{\mathbf{w}_{\mathrm{R}}} \mathcal{L}\left(y \cdot (\mathbf{w}_{\mathrm{R}}^{\mathrm{T}} \mathbf{x}) - \epsilon \|\mathbf{w}_{\mathrm{R}}\|_1\right) \tag{23}$$

Correspondingly, the minimization objective for a baseline model is given by $\min_{\mathbf{w}_{\mathrm{NR}}} \mathcal{L}\left(y \cdot (\mathbf{w}_{\mathrm{NR}}^{\mathrm{T}} \mathbf{x})\right)$. Looking into the training dynamics under gradient descent, we can define the weights at epoch 't' for a baseline and robust model as a function of the Jacobian of the loss function with respect to their corresponding weights, i.e.,

$$\frac{\mathbf{w}_{\mathrm{NR}} - \mathbf{w}_0}{\eta} = \nabla_{\mathbf{w}_{\mathrm{NR}}} \mathcal{L}\left(y . f_{\mathrm{NR}}(\mathbf{x})\right), \tag{24}$$

$$\frac{\mathbf{w}_{\mathrm{R}} - \mathbf{w}_0}{\eta} = \nabla_{\mathbf{w}_{\mathrm{R}}} \mathcal{L}\left(y . f_{\mathrm{R}}(\mathbf{x}) - \epsilon \|\mathbf{w}_{\mathrm{R}}\|_1\right), \tag{25}$$

where $\eta$ is the learning rate of the gradient descent optimizer, $\mathbf{w}_0$ is the weight initialization of both models.

$$\nabla_{\mathbf{w}_{\mathrm{NR}}} \mathcal{L}\left(y . f_{\mathrm{NR}}(\mathbf{x})\right) = -\frac{\exp(-y . f_{\mathrm{NR}}(\mathbf{x}))}{1 + \exp(-y . f_{\mathrm{NR}}(\mathbf{x}))} y . \mathbf{x}^{\mathrm{T}}$$

$$\nabla_{\mathbf{w}_{\mathrm{R}}} \mathcal{L}\left(y . f_{\mathrm{R}}(\mathbf{x}) - \epsilon \|\mathbf{w}_{\mathrm{R}}\|_1\right) = -\frac{\exp(-y . f_{\mathrm{R}}(\mathbf{x}) + \epsilon \|\mathbf{w}_{\mathrm{R}}\|_1)}{1 + \exp(-y . f_{\mathrm{R}}(\mathbf{x}) + \epsilon \|\mathbf{w}_{\mathrm{R}}\|_1)} (-y . \mathbf{x}^{\mathrm{T}} + \epsilon . \mathrm{sign}(\mathbf{w}_{\mathrm{R}})),$$

where $\mathrm{sign}(\mathbf{x})$ return +1, -1, 0 for $x > 0$, $x < 0$, $x = 0$ respectively and $\sigma(\mathbf{x}) = \frac{1}{1 + \exp(-\mathbf{x})}$ is the sigmoid function. Let us denote the weights of the baseline and robust model at the $n$-th iteration as $\mathbf{w}_{\mathrm{NR}}^n$ and $\mathbf{w}_{\mathrm{R}}^n$, respectively. Hence, we can define the $n$-th step of the gradient-descent for both models as:

$$\mathbf{w}_{\mathrm{NR}}^n - \mathbf{w}_{\mathrm{NR}}^{n-1} = \eta \nabla_{\mathbf{w}_{\mathrm{NR}}^{n-1}} \mathcal{L}_{\mathrm{NR}}(\cdot) \tag{26}$$

$$\mathbf{w}_{\mathrm{R}}^n - \mathbf{w}_{\mathrm{R}}^{n-1} = \eta \nabla_{\mathbf{w}_{\mathrm{R}}^{n-1}} \mathcal{L}_{\mathrm{R}}(\cdot), \tag{27}$$

where $\eta$ is the learning rate of the gradient descent optimizer. Taking $n = 1$, we get:

$$\mathbf{w}_{\mathrm{NR}}^1 - \mathbf{w}_{\mathrm{NR}}^0 = \eta \nabla_{\mathbf{w}_{\mathrm{NR}}^0} \mathcal{L}_{\mathrm{NR}}(\cdot) \tag{28}$$

$$\mathbf{w}_{\mathrm{R}}^1 - \mathbf{w}_{\mathrm{R}}^0 = \eta \nabla_{\mathbf{w}_{\mathrm{R}}^0} \mathcal{L}_{\mathrm{R}}(\cdot), \tag{29}$$

where $\mathbf{w}_{\mathrm{NR}}^0$ and $\mathbf{w}_{\mathrm{R}}^0$ are the same initial weights for the baseline and robust models. Subtracting both equations, we get:

$$\frac{\mathbf{w}_{\mathrm{NR}}^1 - \mathbf{w}_{\mathrm{R}}^1}{\eta} = \nabla_{\mathbf{w}_{\mathrm{NR}}^0} \mathcal{L}_{\mathrm{NR}}(\cdot) - \nabla_{\mathbf{w}_{\mathrm{R}}^0} \mathcal{L}_{\mathrm{R}}(\cdot) \tag{30}$$

Similarly, for $n = 2$ and using Equation 30, we get the following relation:

$$\frac{\mathbf{w}_{\mathrm{NR}}^2 - \mathbf{w}_{\mathrm{R}}^2}{\eta} - \left(\frac{\mathbf{w}_{\mathrm{NR}}^1 - \mathbf{w}_{\mathrm{R}}^1}{\eta}\right) = \nabla_{\mathbf{w}_{\mathrm{NR}}^1} \mathcal{L}_{\mathrm{NR}}(\cdot) - \nabla_{\mathbf{w}_{\mathrm{R}}^1} \mathcal{L}_{\mathrm{R}}(\cdot)$$

$$\frac{\mathbf{w}_{\mathrm{NR}}^2 - \mathbf{w}_{\mathrm{R}}^2}{\eta} = \nabla_{\mathbf{w}_{\mathrm{NR}}^0} \mathcal{L}_{\mathrm{NR}}(\cdot) + \nabla_{\mathbf{w}_{\mathrm{NR}}^1} \mathcal{L}_{\mathrm{NR}}(\cdot) - \nabla_{\mathbf{w}_{\mathrm{R}}^0} \mathcal{L}_{\mathrm{R}}(\cdot) - \nabla_{\mathbf{w}_{\mathrm{R}}^1} \mathcal{L}_{\mathrm{R}}(\cdot)$$

Using the above equations, we can now write the difference between the weights of the baseline and robust models at the $n$-th iteration as:

$$\frac{\mathbf{w}_{\text{NR}}^n - \mathbf{w}_{\text{R}}^n}{\eta} = \sum_{i=0}^{n-1} \nabla_{\mathbf{w}_{\text{NR}}^i} \mathcal{L}_{\text{NR}}(\cdot) - \sum_{i=0}^{n-1} \nabla_{\mathbf{w}_{\text{R}}^i} \mathcal{L}_{\text{R}}(\cdot)$$

$$= \sum_{i=0}^{n-1} (\sigma(y.f_{\text{NR}}^i(\mathbf{x})) - 1) y.\mathbf{x}^T - \sum_{i=0}^{n-1} (\sigma(y.f_{\text{R}}^i(\mathbf{x}) - \epsilon||\mathbf{w}_R^i||_1) - 1)(y.\mathbf{x}^T - \epsilon \, \text{sign}(\mathbf{w}_R^i))$$

$$= \sum_{i=0}^{n-1} \sigma(y.f_{\text{NR}}^i(\mathbf{x})) y.\mathbf{x}^T - \sum_{i=0}^{n-1} \Big( \sigma(y.f_{\text{R}}^i(\mathbf{x}) - \epsilon||\mathbf{w}_R^i||_1)(y.\mathbf{x}^T - \epsilon \, \text{sign}(\mathbf{w}_R^i)) + \epsilon \, \text{sign}(\mathbf{w}_R^i) \Big)$$

$$\leq \sum_{i=0}^{n-1} \sigma(y.f_{\text{NR}}^i(\mathbf{x})) y.\mathbf{x}^T - \sum_{i=0}^{n-1} \Big( \sigma(y.f_{\text{R}}^i(\mathbf{x}))(y.\mathbf{x}^T - \epsilon \, \text{sign}(\mathbf{w}_R^i)) + \epsilon \, \text{sign}(\mathbf{w}_R^i) \Big)$$

$$\text{(Using } \sigma(a - b) \leq \sigma(a) \text{ for } b > 0 \text{)}$$

$$\leq \sum_{i=0}^{n-1} \Big( \sigma(y.f_{\text{NR}}^i(\mathbf{x}) - \sigma(y.f_{\text{R}}^i(\mathbf{x})) \Big) y.\mathbf{x}^T + \sum_{i=0}^{n-1} \Big( \sigma(y.f_{\text{R}}^i(\mathbf{x})) - 1 \Big) \epsilon \, \text{sign}(\mathbf{w}_R^i)$$

Using $\ell_2$-norm on both sides, we get:

$$\frac{1}{\eta} \|\mathbf{w}_{\text{NR}}^n - \mathbf{w}_{\text{R}}^n\|_2 \leq \| \sum_{i=0}^{n-1} \Big( \sigma(y.f_{\text{NR}}^i(\mathbf{x}) - \sigma(y.f_{\text{R}}^i(\mathbf{x})) \Big) y.\mathbf{x}^T + \sum_{i=0}^{n-1} \Big( \sigma(y.f_{\text{R}}^i(\mathbf{x})) - 1 \Big) \epsilon \, \text{sign}(\mathbf{w}_R^i) \|_2$$

$$\leq \| \sum_{i=0}^{n-1} \Big( \sigma(y.f_{\text{NR}}^i(\mathbf{x}) - \sigma(y.f_{\text{R}}^i(\mathbf{x})) \Big) y.\mathbf{x}^T \|_2 + \| \sum_{i=0}^{n-1} \Big( \sigma(y.f_{\text{R}}^i(\mathbf{x})) - 1 \Big) \epsilon \, \text{sign}(\mathbf{w}_R^i) \|_2$$

$$\text{(Using Triangle Inequality)}$$

$$\leq \| \sum_{i=0}^{n-1} \Big( \sigma(y.f_{\text{NR}}^i(\mathbf{x}) - \sigma(y.f_{\text{R}}^i(\mathbf{x})) \Big) y.\mathbf{x}^T \|_2 + \epsilon\sqrt{d} \sum_{i=0}^{n-1} \|\sigma(y.f_{\text{R}}^i(\mathbf{x})) - 1\|_2$$

$$\leq \sum_{i=0}^{n-1} \| \Big( \sigma(y.f_{\text{NR}}^i(\mathbf{x}) - \sigma(y.f_{\text{R}}^i(\mathbf{x})) \Big) y.\mathbf{x}^T \|_2 + \epsilon\sqrt{d} \sum_{i=0}^{n-1} \|\sigma(y.f_{\text{R}}^i(\mathbf{x})) - 1\|_2$$

$$\leq \sum_{i=0}^{n-1} \| \Big( \sigma(y.f_{\text{NR}}^i(\mathbf{x})) - \sigma(y.f_{\text{R}}^i(\mathbf{x})) \Big) \|_2 \|y.\mathbf{x}^T\|_2 + \epsilon\sqrt{d} \sum_{i=0}^{n-1} \|\sigma(y.f_{\text{R}}^i(\mathbf{x})) - 1\|_2$$

$$\leq n\|y.\mathbf{x}^T\|_2 + \epsilon\sqrt{d} \sum_{i=0}^{n-1} \|\sigma(y.f_{\text{R}}^i(\mathbf{x})) - 1\|_2$$

$$\leq n\|y.\mathbf{x}^T\|_2 + \epsilon\sqrt{d} \sum_{i=0}^{n-1} (1 - \sigma(y.f_{\text{R}}^i(\mathbf{x}))) \quad \text{(since the term inside } \|\cdot\|_2 \text{ is a scalar)}$$

$$\leq n\|y.\mathbf{x}^T\|_2 + \epsilon\sqrt{d}n$$

$$\|\Delta_{\mathbf{w}}\|_2 \leq n\eta(y\|\mathbf{x}^T\|_2 + \epsilon\sqrt{d})$$

$$\square$$

## A.4 VALIDITY

**Theorem 3.** *(Validity Comparison) For a given instance* $\mathbf{x} \in \mathbb{R}^d$ *and desired target label denoted by unity, let* $\mathbf{x}_{\text{R}}$ *and* $\mathbf{x}_{\text{NR}}$ *be the counterfactuals for adversarially robust* $f_{\text{R}}(\mathbf{x})$ *and non-robust* $f_{\text{NR}}(\mathbf{x})$ *models respectively. Then,* $Pr(f_{\text{NR}}(\mathbf{x}_{\text{NR}}) = 1) \geq Pr(f_{\text{R}}(\mathbf{x}_{\text{R}}) = 1)$ *if* $|f_{\text{NR}}(\mathbf{x}_{\text{R}}) - f_{\text{NR}}(\mathbf{x}_{\text{NR}})| \leq n\eta(y\|\mathbf{x}^T\|_2 + \epsilon\sqrt{d})\|\mathbf{x}_{\text{R}}\|$, *where* $\eta$ *is the learning rate,* $\epsilon$ *is the* $\ell_2$*-norm perturbation ball,* $y$ *is the label for* $\mathbf{x}$, *and* $n$ *is the total number of training epochs.*

*Proof.* In a logistic regression case, $Pr(f(\mathbf{x}) = 1) = \frac{e^{\mathbf{w}^T\mathbf{x}}}{1+e^{\mathbf{w}^T\mathbf{x}}}$, which is the sigmoid of the model output. Next, we derive the difference in probability of a valid recourse from non-robust and adversarially robust model:

$$Pr(f_{\text{NR}}(\mathbf{x}_{\text{NR}}) = 1) - Pr(f_{\text{R}}(\mathbf{x}_{\text{R}}) = 1) = \frac{e^{\mathbf{w}_{\text{NR}}^T\mathbf{x}_{\text{NR}}}}{1 + e^{\mathbf{w}_{\text{NR}}^T\mathbf{x}_{\text{NR}}}} - \frac{e^{\mathbf{w}_{\text{R}}^T\mathbf{x}_{\text{R}}}}{1 + e^{\mathbf{w}_{\text{R}}^T\mathbf{x}_{\text{R}}}} \tag{31}$$

$$= \frac{e^{\mathbf{w}_{\text{NR}}^T\mathbf{x}_{\text{NR}}} - e^{\mathbf{w}_{\text{R}}^T\mathbf{x}_{\text{R}}}}{(1 + e^{\mathbf{w}_{\text{R}}^T\mathbf{x}_{\text{R}}})(1 + e^{\mathbf{w}_{\text{NR}}^T\mathbf{x}_{\text{NR}}})} \tag{32}$$

Since $(1 + e^{\mathbf{w}_{\text{R}}^T\mathbf{x}_{\text{R}}})(1 + e^{\mathbf{w}_{\text{NR}}^T\mathbf{x}_{\text{NR}}}) > 0$, so $Pr(f_{\text{NR}}(\mathbf{x}_{\text{NR}}) = 1) \geq Pr(f_{\text{R}}(\mathbf{x}_{\text{R}}) = 1))$ occurs when,

$$e^{\mathbf{w}_{\text{NR}}^T\mathbf{x}_{\text{NR}}} \geq e^{\mathbf{w}_{\text{R}}^T\mathbf{x}_{\text{R}}} \tag{33}$$

$$\mathbf{w}_{\text{NR}}^T(\mathbf{x}_{\text{NR}} - \mathbf{x}_{\text{R}}) \geq (\mathbf{w}_{\text{R}}^T - \mathbf{w}_{\text{NR}}^T)\mathbf{x}_{\text{R}} \qquad \text{(Taking natural logarithm on both sides)}$$

$$\mathbf{w}_{\text{NR}}^T(\mathbf{x}_{\text{R}} - \mathbf{x}_{\text{NR}}) \leq (\mathbf{w}_{\text{NR}}^T - \mathbf{w}_{\text{R}}^T)\mathbf{x}_{\text{R}} \tag{34}$$

$$\left\| \mathbf{w}_{\text{NR}}^T(\mathbf{x}_{\text{R}} - \mathbf{x}_{\text{NR}}) \right\| \leq \left\| (\mathbf{w}_{\text{NR}}^T - \mathbf{w}_{\text{R}}^T)\mathbf{x}_{\text{R}} \right\| \qquad \text{(Taking norm on both sides)}$$

$$\left\| \mathbf{w}_{\text{NR}}^T(\mathbf{x}_{\text{R}} - \mathbf{x}_{\text{NR}}) \right\| \leq \left\| \mathbf{w}_{\text{NR}} - \mathbf{w}_{\text{R}} \right\| \|\mathbf{x}_{\text{R}}\| \qquad \text{(Using Cauchy-Schwartz)}$$

$$\left\| \mathbf{w}_{\text{NR}}^T(\mathbf{x}_{\text{R}} - \mathbf{x}_{\text{NR}}) \right\| \leq n\eta(y\|\mathbf{x}^T\|_2 + \epsilon\sqrt{d})\|\mathbf{x}_{\text{R}}\| \qquad \text{(From Lemma 1)}$$

$$|f_{\text{NR}}(\mathbf{x}_{\text{R}}) - f_{\text{NR}}(\mathbf{x}_{\text{NR}})| \leq n\eta(y\|\mathbf{x}^T\|_2 + \epsilon\sqrt{d})\|\mathbf{x}_{\text{R}}\| \tag{35}$$

$\square$

## B    ADDITIONAL EXPERIMENTAL RESULTS

In this section, we have plots for cost differences, validity, and adversarial accuracy for the two logistic regression and neural network models trained on three real-world datasets.

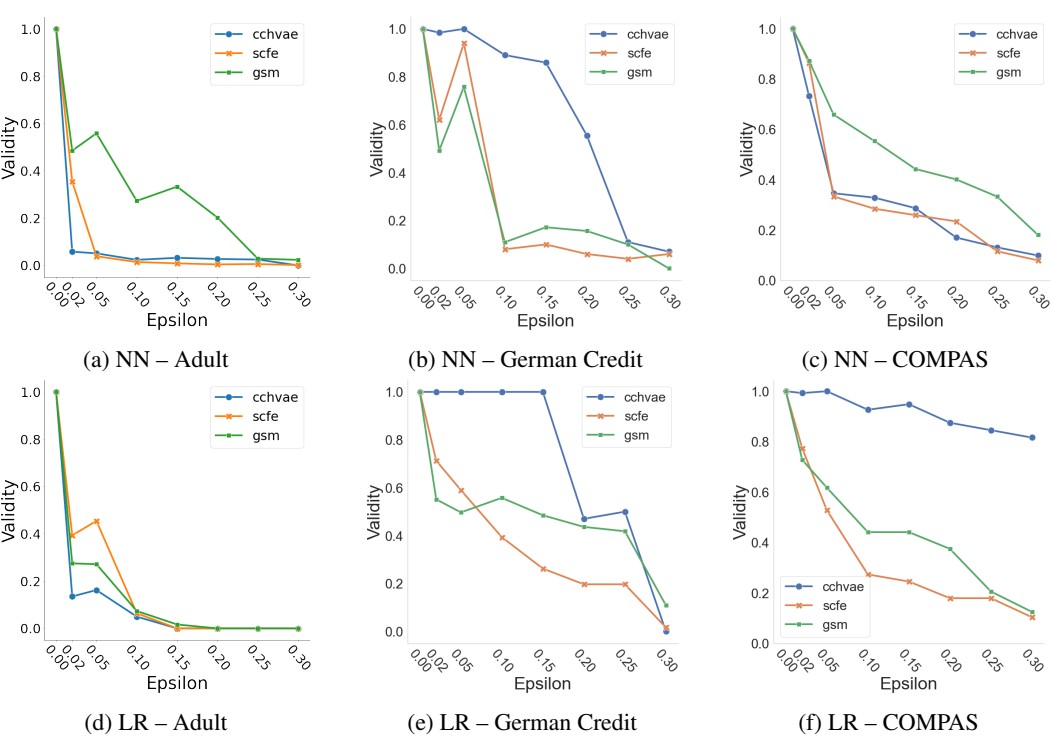

Figure 5: Analyzing validity of recourse generated using non-robust and adversarially robust Logistic Regression(LR) and Neural Networks (NN) for Adult, COMPAS, and German Credit datasets. We find that the validity decreases for increasing values of $\epsilon$.

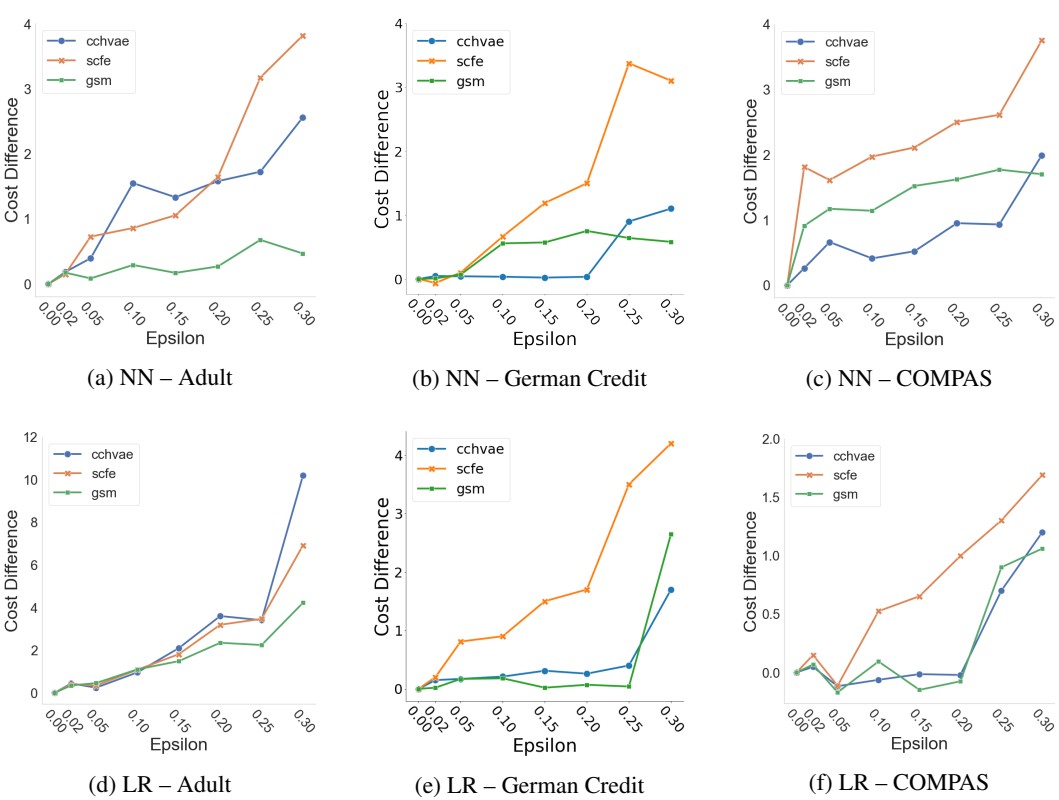

Figure 6: Analyzing cost differences between recourse generated using non-robust and adversarially robust Logistic Regression (LR) and Neural Networks(NN) for Adult, COMPAS, and German Credit datasets. We find that the cost difference (i.e., $\ell_2-$norm) between the recourses generated for non-robust and adversarially robust models increases for increasing values of $\epsilon$.

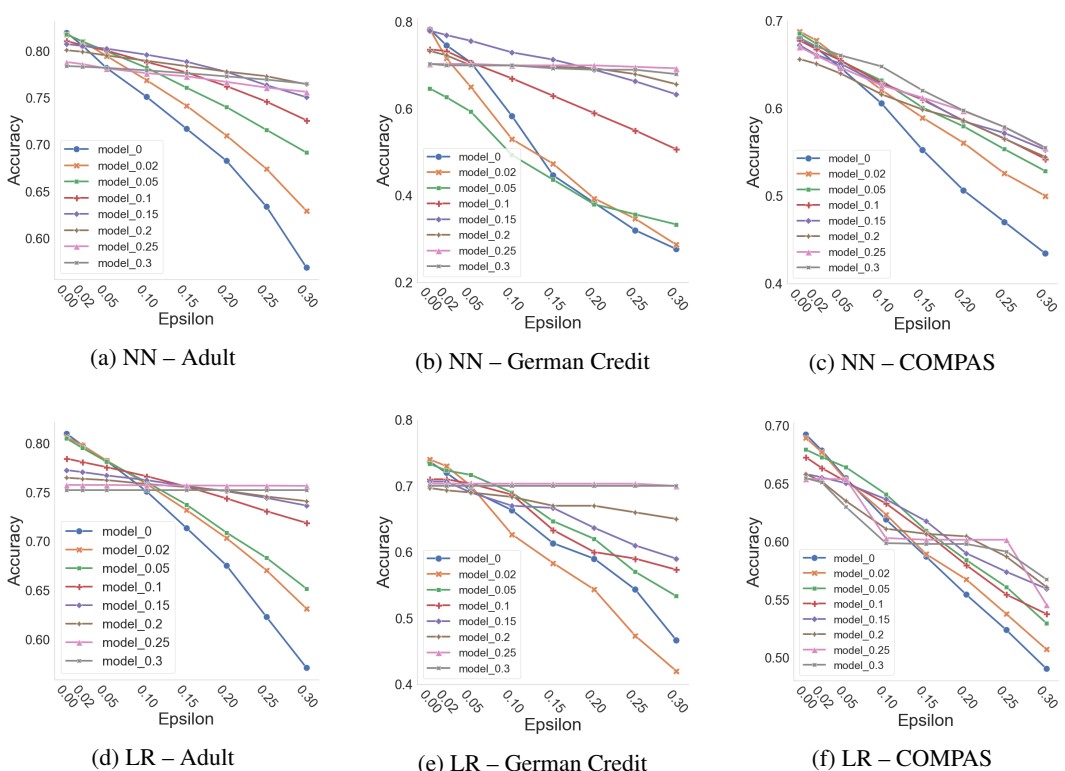

Figure 7: Here we plot the adversarial accuracy of the different models we trained on varying degree of robustness ($\epsilon$). As expected, we observe the adversarial accuracy for the non-robust model is lowest out of all, and gradually gets better when the model is adversarially trained.

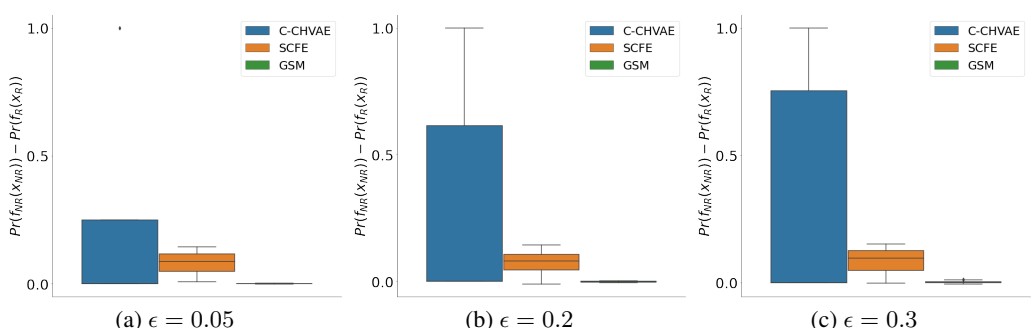

Figure 8: Comparison between the validity of recourses generated for non-robust and adversarially robust model for varying degrees of robustness ($\epsilon$)

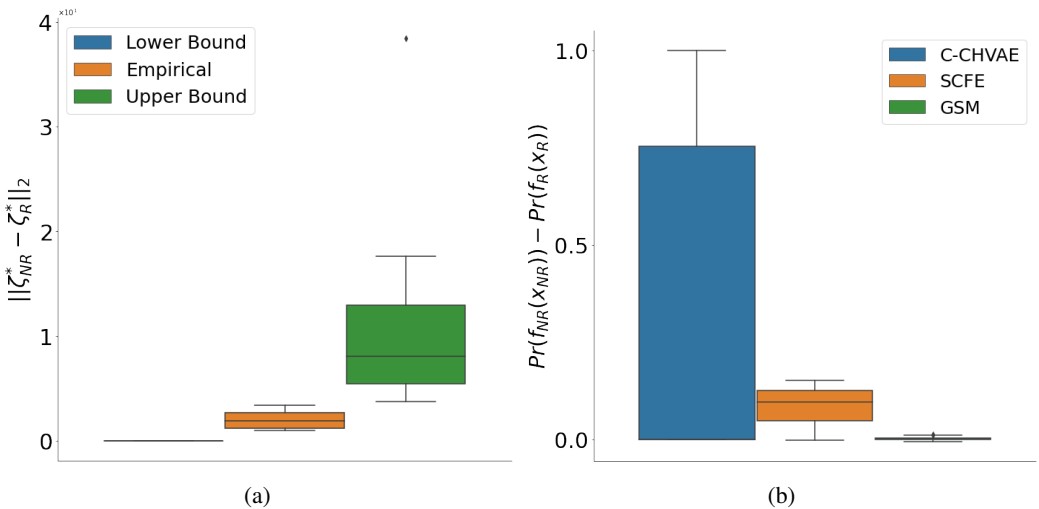

(a)                                  (b)

Figure 9: (Linear Approximation Bound Verification) (a) Empirically calculated cost differences (in orange) for the original model and our theoretical lower (in blue) and upper (in green) bounds for SCFE recourses corresponding to adversarially robust (trained using $\epsilon=0.3$) vs. non-robust linear approximation of neural networks corresponding to test samples of the Adult dataset. Figure (b) is the empirical difference between the validity of recourses for non-robust and adversarially robust linear approximated model. Results show no violations of our theoretical bounds. See Appendix B for results using different $\epsilon$ values.

