# OpenReview forum: "On the Impact of Adversarially Robust Models on Algorithmic Recourse"
_ICLR.cc/2023/Conference — Submitted to ICLR 2023_

### Official Review · Reviewer_aAvX · 2022-10-23

**Confidence:** 3
**Correctness:** 3
**Technical Novelty And Significance:** 3
**Empirical Novelty And Significance:** 3
**Recommendation:** 3

**Clarity, Quality, Novelty And Reproducibility:**

The paper is clearly written and the quality is good. My major concern is about the lack of novelty, given the exising work [1].

**Strength And Weaknesses:**

Strength:

The structure of the paper is clear. Both theoretical analyses and empirical study are carried out to study the connection between adversarial robustness and algorithmic recourse.


Weaknesses:

The contribution is not significant, since the connection between adversarial robustness and algorithmic recourse has been unveiled by a previous paper [1]. It seems that this submission is a simple extension of [1]: proving the upper and lower bound for cost distance assuming a linear model and running the experiment on the same datasets.

More importantly, I cannot agree with the argument that improved adversarial robustness leads to harder recourse for a general deep neural network. If the features are disentangled and model is linear, the adversarial robustness is only related to the $l_p$ norm of weight, where $p$ depends on the attack budget norm. I guess the mentioned trade-off in this paper is related to the $l_p$ norm of weight: if the norm is large, the robustness is weak but the attack (recourse) is simple; if the norm is small, the opposite holds. But this logic is possibly not true for deep neural nets, where the features are entangled and the model is highly nonlinear. Adversarial training for a deep NN aims to purify the features so that the features are correlated with the target labels. This does not conflict with algorithmic recourse, since the adversarial robustness helps learn a meaningful feature space. If the paper only cares the simple case, please clarify it at the abstract or introduction. But if this paper only studies a simple linear model, I cannot see a major contribution here compared with [1].

[1] Martin Pawelczyk, Chirag Agarwal, Shalmali Joshi, Sohini Upadhyay, and Himabindu Lakkaraju. Exploring counterfactual explanations through the lens of adversarial examples: A theoretical and empirical analysis. In International Conference on Artificial Intelligence and Statistics (AISTATS)


1. All of the theoretical analysis seem to assume the model is a linear one. So I think there is a gap between the analysis and empirical study, where a neural network is used to show the correctness of the bound.

2. Does the assumption in Theorem 3 generally hold in the real datasets?

3. Some references are duplicates:

Martin Pawelczyk, Klaus Broelemann, and Gjergji Kasneci. Learning model-agnostic counterfactual explanations for tabular data. In Proceedings of The Web Conference 2020 (WWW). ACM, 2020a.

Martin Pawelczyk, Klaus Broelemann, and Gjergji Kasneci. Learning model-agnostic counterfactual explanations for tabular data. In Proceedings of The Web Conference 2020, pp. 3126–3132, 2020b.

Martin Pawelczyk, Chirag Agarwal, Shalmali Joshi, Sohini Upadhyay, and Himabindu Lakkaraju. Exploring counterfactual explanations through the lens of adversarial examples: A theoretical and empirical analysis. In International Conference on Artificial Intelligence and Statistics (AISTATS), 2022a.

Martin Pawelczyk, Chirag Agarwal, Shalmali Joshi, Sohini Upadhyay, and Himabindu Lakkaraju. Exploring counterfactual explanations through the lens of adversarial examples: A theoretical and empirical analysis. In AISTATS. PMLR, 2022b.


**Summary Of The Paper:**

This paper studies the cost and validity of algorithmic recourse when the model is adversarially robust. It first shows upper and lower bounds for cost and a theorem showing that the validity of non-robust model is higher than that of a robust counterpart. Then the paper shows the empirical result on three datasets, which shows the correctness of the bounds and validity theorem.

**Summary Of The Review:**

I think the major contribution of this major is not substantial given the previous paper [1], so I give a negative score at this phase.

---

> ### Author Response · Authors · 2022-11-18
> **Response to Reviewer aAvX**
>
> We thank the reviewer for their insightful comments, and acknowledging the theoretical and empirical analysis to study the impact of a model’s adversarial robustness on algorithmic recourse. Below, we address specific concerns raised by the reviewer.
>
>
> **“The contribution is not significant, since the connection between adversarial robustness and algorithmic recourse has been unveiled by a previous paper [1].”**
>
> There is a significant difference between the two works. [1] shows that adversarial examples and counterfactual examples are not very far from each other for a given model, supported by theoretical and empirical analysis. Whereas our work provides the analysis about the impact on finding a counterfactual when the model is adversarially robust. Hence, the goal of the two works are very different.
>
>
> **“But if this paper only studies a simple linear model, I cannot see a major contribution here compared with [1]”**
>
> We would like to clarify that we do not assume anything about the entanglement of the features. While we do make linearity and exponential family assumptions in our theoretical analysis, this is an accepted practice in the recourse literature. For example, the theoretical analysis in Ustun et al.’s [2] seminal paper makes linear model assumptions as well. In addition, Ustun et al. [2] argue that their approach and the accompanying theory (which relies on the linear model assumption) can be readily applied to non-linear models by first generating local linear approximations of any given non-linear model using algorithms such as LIME (Ribeiro et. al.[3]). This is also supported by our results on non-linear models (neural networks) where cost difference and validity follow the same trend as observed for linear models  (see Section 5.2 for more details). In order to justify that our theoretical analysis holds for linear approximation of the non-linear models, we plotted theoretical upper and lower bounds for linear approximation of the neural network and observe that the bounds hold for the linear approximation of the neural network corresponding to all the test samples of the Adult dataset (ref Figure 9).
>
> **”Does the assumption in Theorem 3 generally hold in the real datasets?”**
>
> Yes, the assumption holds for at least 90% in all the datasets (also added to the updated version in Section 5.2 under “Empirical Verification of Theoretical Bounds”).
>
>
>
> **References**
>
> - [1] Pawelczyk, Martin, et al. "Exploring counterfactual explanations through the lens of adversarial examples: A theoretical and empirical analysis." International Conference on Artificial Intelligence and Statistics. PMLR, 2022.
>
> - [2] Ustun, Berk, Alexander Spangher, and Yang Liu. "Actionable recourse in linear classification." Proceedings of the conference on fairness, accountability, and transparency. 2019.
>
> - [3] Ribeiro, Marco Tulio, Sameer Singh, and Carlos Guestrin. "" Why should i trust you?" Explaining the predictions of any classifier." Proceedings of the 22nd ACM SIGKDD international conference on knowledge discovery and data mining. 2016.

---

> > ### Comment · Reviewer_aAvX · 2022-12-02
> > **Follow-up**
> >
> > Thanks for explaining the difference between [1] and this submission. However, I cannot agree there is significant difference. Thus, I will keep my original score.

---

### Official Review · Reviewer_G7SB · 2022-10-24

**Confidence:** 3
**Correctness:** 1
**Technical Novelty And Significance:** 2
**Empirical Novelty And Significance:** 2
**Recommendation:** 3

**Clarity, Quality, Novelty And Reproducibility:**

Clarity: This paper is in general not clearly written. There are variables used without explanation. For example, what is $\boldsymbol{w}$ in Eq. (5)?

Quality: There are serious errors in this paper. Please see Weakness W2) for example.

Novelty: It should be recognized as novel if the theoretical results are correct. However, there are errors in the proofs of the bounds.

Reproducibility: There are so many details in the experiments and it seems not easily reproduced without shared codes.

**Strength And Weaknesses:**

Strength:

S1) Effects of adversarial training on algorithmic resource are analyzed by establishing bounds on the differences in cost and validity performance of the non-robust and adversarially robust models.

S2) The proposed error bounds are verified by numerical experiments.

Weakness:

W1) The analyzed classification model $f(\boldsymbol{x})=\phi(h(\boldsymbol{x}))$ seems somewhat simple, and may not well reflect the properties of modern deep models.

W2) There may be serious errors in the proof.

For example, in Page 13, the authors use the following inequations
$$\zeta^*_{NR}\ge (s-\boldsymbol{w}\_{NR}^{T}\boldsymbol{x} )\frac{\lambda}{\lambda+1}\boldsymbol{w}\_{NR}$$
and
$$\zeta^*_{R} \ge  (s-\boldsymbol{w}^{T}\_{R}\boldsymbol{x} )\frac{\lambda}{\lambda+1}\boldsymbol{w}\_{R}$$
to obtain the inequality in Line 15 as follows:
$$\zeta^*_{NR}-\zeta^*_{R} \ge \frac{\lambda}{\lambda+1}\big(  (s-\boldsymbol{w}^{T}\_{NR}\boldsymbol{x} )\boldsymbol{w}\_{NR} - (s-\boldsymbol{w}^{T}\_{R}\boldsymbol{x} )\boldsymbol{w}\_{R}\big)$$


However, this derivation may not be correct because we cannot directly get $a-b\ge x-y$ from $a\ge x$ and $b\ge y$. Did I miss something?

(After rebuttal: The authors have conducted a careful revision and fixed this serious error in the proof which of course changed the original theorem. )

**Summary Of The Paper:**

This paper studies the effect of adversarial training on the algorithmic recourse  by deriving both cost bounds and validity bounds. The correctness of the proposed bounds are verified through numerical simulation.

**Summary Of The Review:**

Before rebuttal: Due to the comments in "Clarity, Quality, Novelty And Reproducibility" especially serious errors in the proof, I recommend "Strong reject".

After rebuttal: The authors have conducted a careful revision and fixed a serious error in the proof. However, the novelty as well as the writing seems still not so satisfactory. As a result, I updated my score from "1: strong reject" to "3: reject".

---

> ### Author Response · Authors · 2022-11-18
> **Response to Reviewer G7SB**
>
> We thank the reviewer for their insightful comments, and acknowledging the novelty of our analysis in studying the impact of adversarial robust models on the cost and validity of the recourse. Below, we address specific concerns raised by the reviewer.
>
> **“W1) The analyzed classification model f(x)=ϕ(h(x)) seems somewhat simple, and may not well reflect the properties of modern deep models”**
>
> While we do make linearity and exponential family assumptions in our theoretical analysis, this is an accepted practice in the recourse literature. For example, the theoretical analysis in Ustun et al.’s [3] seminal paper makes linear model assumptions as well.
>
> Furthermore, we believe that these assumptions do not narrow down the scope of our results because recent literature in algorithmic recourse commonly considers local linear approximations of non-linear models. For instance, Ustun et al. [3] argue that their approach and the accompanying theory (which relies on the linear model assumption) can be readily applied to non-linear models by first generating local linear approximations of any given non-linear model using algorithms such as LIME (Ribeiro et. al.[4]). This is also supported by our results on non-linear models (neural networks) where cost difference and validity follow the same trend as observed for linear models  (see Section 5.2 for more details).
>
> **“W2) There may be serious errors in the proof.”**
>
> We have added the underlying assumption needed to fully understand the proof, and while adding it we found a more concise proof which also provides us with tighter lower bounds on the difference between the cost of finding recourse for non-robust model and robust model.
>
> **“What is w in Eq. (5)”**
>
> We apologize for the confusion and would like to clarify that the $w$ in Equation 5 refers to the weights of the linear model. In response to the reviewers’ comment, we have modified the description of $w$  in Eq. (5) the revised version of the manuscript for more clarity.
>
> **Quality**
> We addressed the concerns about the potential inconsistency in the proof above under “Response to W2”.
>
> **Novelty**
>
> Thanks for acknowledging the novelty of our work. We addressed the concerns about the potential inconsistency in the proof above under “Response to W2”.
>
> **Reproducibility**
>
> We apologize for not providing the code in the initial submission. We have uploaded the code for our work in  this revision.
>
>
> **References**
>
> - [1] Ribeiro, Marco Tulio, Sameer Singh, and Carlos Guestrin. “Why should i trust you?" Explaining the predictions of any classifier." Proceedings of the 22nd ACM SIGKDD international conference on knowledge discovery and data mining. 2016.
> - [2] Martin Pawelczyk, Chirag Agarwal, Shalmali Joshi, Sohini Upadhyay, and Himabindu Lakkaraju. Exploring counterfactual explanations through the lens of adversarial examples: A theoretical and empirical analysis. In International Conference on Artificial Intelligence and Statistics (AISTATS)
> - [3] Ustun, Berk, Alexander Spangher, and Yang Liu. "Actionable recourse in linear classification." Proceedings of the conference on fairness, accountability, and transparency. 2019.
> - [4] Ribeiro, Marco Tulio, Sameer Singh, and Carlos Guestrin. "" Why should i trust you?" Explaining the predictions of any classifier." Proceedings of the 22nd ACM SIGKDD international conference on knowledge discovery and data mining. 2016.

---

> > ### Comment · Reviewer_G7SB · 2022-11-19
> > **Response to the authors' feedback**
> >
> > Many thanks for the authors' feedback. The authors have conducted a careful revision and fixed a serious error in the proof. However, the novelty as well as the writing seems still not so satisfactory. As a result, I updated my score from "1: strong reject" to "3: reject".

---

### Official Review · Reviewer_em4b · 2022-10-24

**Confidence:** 4
**Correctness:** 3
**Technical Novelty And Significance:** 3
**Empirical Novelty And Significance:** 3
**Recommendation:** 6

**Clarity, Quality, Novelty And Reproducibility:**

The organization and presentation of the paper is excellent. It was a pleasure reading the paper.

Can you please check your references, there appears to be quite a few repetitions

Pawelczyk 2020a and 2020b

Pawelczyk 2022a and 2022b

Upadhyay 2021a and 2021b

Ustun 2019a and 2019b

Minor typos
Page 5 - “of the weights $w_R and w_B$ - $w_{NR}$?
Page 7 - Evaluation metrics, cost definition perhaps is missing a summation on the test data points? Similarly the validity definition is also missing a summation on test data points.

Figure 1(b), the legend is obscuring the plot for the upper bound.


**Strength And Weaknesses:**

Strengths

It is obvious that ensuring adversarial robustness in models will have a bearing on algorithmic recourse techniques, seemingly making it harder to find valid recourses. The paper does a good job of theoretically analyzing this aspect. It does not propose a new technique, rather analyses the impact of a model’s adversarial robustness on algorithmic recourse. The accompanying experiments are exhaustive and validate the bounds.


Weaknesses

The authors mention that categorical features are removed for efficient training of their models. It is unclear why is this so? Excluding a certain category of features that are typical in datasets for studying algorithmic recourse is not a good idea. If the adversarial robustness algorithm cannot handle categorical features, then it is a major limitation of the analysis.

There are recourse generation techniques that are not gradient descent/ascent based and involve discrete optimization (to manage categorical attributes). A discussion on how adversarial robustness impacts these algorithms will make the paper more comprehensive.

Disclaimer: I did not carefully check the proofs in the appendix. I only glanced through them.


**Summary Of The Paper:**

Algorithmic recourse techniques search for a valid recourse in the vicinity of a point, while adversarial robustness ensures that the model outputs do not change in the vicinity of a point! Algorithmic recourse techniques for models trained to ensure adversarial robustness have a larger work to do than the non-robust counterparts. This paper aims at understanding just how difficult and different are the recourses generated for models that are adversarially robust. The authors derive theoretical (lower and upper) bounds on the differences in the reccourses generated by popular recourse generation techniques for adversarial robust models and their non-robust counterparts. These bounds are empirically tested using three datasets popular in the algorithmic recourse community.


**Summary Of The Review:**

Overall, the authors do a thorough job of connecting two similar (yet with seemingly conflicting objectives) lines of work - algorithmic recourse and adversarial robustness. The theoretical bounds on differences in the recourses generated on models that are adversarially robust and their non-robust counterparts, and their thorough empirical validation is a good contribution to the algorithmic recourse community.

---

> ### Author Response · Authors · 2022-11-18
> **Response to Reviewer em4b**
>
> We thank the reviewer for their insightful comments, and acknowledging the comprehensive theoretical and empirical analysis to study the impact of a model’s adversarial robustness on algorithmic recourse.. Below, we address specific concerns raised by the reviewer.
>
> **“Excluding a certain category of features that are typical in datasets for studying algorithmic recourse is not a good idea.”**
>
> As mentioned in [1, 2], the reason behind removing categorical features is because adversarial robustness methods are designed to provide endurance against adversarial examples, which by definition are imperceptible changes to the input sample. However, this imperceptible change is not valid in categorical features since changing categorical features is in fact highly visible. For instance, changing gender feature value from “male” to “female” cannot be considered an imperceptible change. Hence, most existing works related to adversarial robustness for tabular data including ours consider only continuous features. This was also the reason behind not focussing on discrete optimization recourse algorithms specifically since it would not provide a clear picture of the impact of adversarial robust models which are primarily focussed on adversarial examples.
>
>
> **References**
>
> - [1] Erdemir, Ecenaz, et al. "Adversarial robustness with non-uniform perturbations." Advances in Neural Information Processing Systems 34 (2021): 19147-19159.
> - [2] Ballet, Vincent, et al. "Imperceptible adversarial attacks on tabular data." arXiv preprint arXiv:1911.03274 (2019).

---

> > ### Comment · Reviewer_em4b · 2022-11-20
> > **response to authors' feedback**
> >
> > I thank the authors for clarifying the issue with categorical attributes. After reading the other comments, I am inclined to keep my original rating.

---

### Official Review · Reviewer_T2PC · 2022-10-25

**Confidence:** 3
**Correctness:** 2
**Technical Novelty And Significance:** 2
**Empirical Novelty And Significance:** 2
**Recommendation:** 3

**Clarity, Quality, Novelty And Reproducibility:**

This paper is easy to follow, while assumptions should be explicitly stated in lemmas and theorems.

**Strength And Weaknesses:**

Strengths:

- It is important to study the trade-off between the model's adversarial robustness and the feasibility of algorithmic recourse. Adversarial robustness pursues the cost of adversarial examples to be high, while algorithmic recourse searches low-cost modifications. There seems to be an inherent trade-off between the two goals.
- Experiments are thorough.

Weaknesses:

- All of the theoretical results in the paper essentially rely on the assumption that the considered classifier is linear. This assumption is important, and thus should be explicitly stated. For example, in Definition 1, it is not enough to write "$f(x)$ is a local linear score approximation", in which $f(x)=w^Tx+b$ should be explicitly stated, as Pawelczyk et al did.
- Lemma 1 and Theorem 3 also heavily rely on the assumption that the classifier is linear. The assumption should be explicitly written in the statement.
- Moreover, Lemma 1 relies on the assumption that the adversarially robust model is trained to minimize the worst-case loss with $\ell_{\infty}$ norm perturbation. This assumption should also be explicitly stated in the main text.
- Similarly, Definition 1 relies on another assumption that SCFE is generated using $\ell_2$ norm as the distance metric. This point should also be explicitly stated. Otherwise, readers have to look for the details of this definition in Pawelczyk et al, 2022.
- After I notice the above implicit assumption by myself, an important yet secret issue surfaced. That is, the norm used to measure adversarial robustness and the norm used to generate algorithmic recourse are not consistent. This inconsistency is unpleasant, because the norm defines what adversarial robustness really means. It would be misleading to draw rough conclusions based on the relationship between $\ell_{\infty}$ robustness and $\ell_2$-based recourse. Before that, the relationship between $\ell_{2}$ robustness and $\ell_2$-based recourse should be studied first.

**Summary Of The Paper:**

This paper studies whether adversarially robust models will provide algorithmic recourses with higher costs than normal models. Evidently, the prediction of adversarially robust models is more robust than normal models under input perturbations, so changing the prediction will take a higher cost. The authors present a theoretical analysis of the cost and validity difference between robust and normal models, based on linear classifiers. Both theoretical and experimental results show model robustness can harm the feasibility of algorithmic recourse.

**Summary Of The Review:**

The problem studied in the paper (i.e., the trade-off between adversarial robustness and algorithmic recourse) is worth exploring. However, many assumptions used by the theoretical analysis are not explicitly stated in the lemmas and theorems. More importantly, in this paper, the metrics used by adversarial training and algorithmic recourse are inconsistent, and this inconsistency is not addressed or discussed.

---

> ### Author Response · Authors · 2022-11-18
> **Response to Reviewer T2PC**
>
> We thank the reviewer for their insightful comments, and for acknowledging the significance of our contributions. Below, we address specific concerns raised by the reviewer.
>
> **“..essentially rely on the assumption that the considered classifier is linear.”, “The assumption should be explicitly written in the statement.”**
>
> Our analysis is applicable to linear models or local linear approximations of complex models (e.g., deep neural networks). Note that considering local linear approximations is common in recourse literature. For example, the theoretical analysis in Ustun et al.’s [2] seminal paper makes linear model assumptions as well. ,we have added more emphasis towards this assumption for better clarity. Furthermore, we moved most of the mathematical assumptions such as the value of p of L-p norms from the appendix to the theoretical section (Section 3,4) for better readability.
>
>
>
>
>
>
> **“It would be misleading to draw rough conclusions based on the relationship between ℓ∞ robustness and ℓ2-based recourse.”**
>
> While the reviewer is correct, there are two important reasons of using current configuration of norms : (1) SCFE has commonly been used with L2 norms [1] while adversarial robustness is applied on linear models with L_inf norm due to the resulting closed form solution [2], (2) SCFE is not analytically solvable with L_inf norm since it makes the objective function non-differentiable resulting in no closed-form solution. Hence, this configuration gives us the best possible combination to analyze cost and validity theoretically.
>
>
> **References**
>
> - [1] Pawelczyk, Martin, et al. "Exploring counterfactual explanations through the lens of adversarial examples: A theoretical and empirical analysis." International Conference on Artificial Intelligence and Statistics. PMLR, 2022.
> - [2] Ustun, Berk, Alexander Spangher, and Yang Liu. "Actionable recourse in linear classification." Proceedings of the conference on fairness, accountability, and transparency. 2019.

---

> > ### Comment · Reviewer_T2PC · 2022-12-02
> > **Thanks for the rebuttal**
> >
> > Thanks for the authors' response. I am still concerned about the inconsistency of norms. If the closed-form solution cannot be found in previous work, I thought it would be better for the authors to address the challenge, instead of evading, especially for the case that the inconsistency significantly weakens the generality of the main conclusion. Therefore, I decided to retain my score.

---

### Decision · Program_Chairs · 2023-01-20

**Decision:**

Reject

**Justification For Why Not Higher Score:**

Despite engagement with the authors, the reviewers continued to argue for rejection, seeing the rebuttals and counterarguments as unconvincing. The main issues centered around novelty, correctness, and validity (norm issue).

**Justification For Why Not Lower Score:**

N/A

**Metareview: Summary, Strengths And Weaknesses:**


This work studies the interplay between algorithmic recourse and adversarial robustness. This is an important problem. Using an analysis centered around a linear modeling assumption, they find that adversarially robust models increase the cost and lower the validity of recourses. This is an important result.

However, several aspects of the paper caught the attention of reviewers. Three out of four reviewers argued for rejection; one argued for weak acceptance.

One reviewer latched on to the discrepancy between the norms used in recourse and in robustness (l_2 v. l_infinity). The reviewers motivated this choice on the grounds of expediency. The reviewer was unconvinced. Another reviewer latched on to the relationship with Pawelczyk et al, and saw this work as highly derivative and not sufficiently novel. I personally found the author's rebuttal to lay out reasonable arguments explaining how these two pieces of work differed, but the reviewer was unmoved. Yet another reviewer found a serious error in the proof, the authors posted a correciton, but then the reviewer decided to stick with their score regardless.

It seems to me that revisions that clearly lay out the relationship with Pawelczyk et al, clearly motivate the choice of norms (on scientific rather than expediency grounds), and that are free of errors in proofs will be much better received by the next set of reviewers to meet this work.

**Summary Of Ac-Reviewer Meeting:**

N/A